# Transductive Decoupled Variational Inference for Few-Shot Classification

**Anuj Singh** [1,2], **Hadi Jamali-Rad** [1,2]

*{a.r.singh, h.jamalirad}@tudelft.nl, {anuj.singh2, hadi.jamali-rad}@shell.com*
[1] *Delft University of Technology, The Netherlands*
[2] *Shell Global Solutions International B.V., Amsterdam, The Netherlands*

**Reviewed on OpenReview:** *https://openreview.net/forum?id=bomdTc9HyL*

## Abstract

The versatility to learn from a handful of samples is the hallmark of human intelligence. Few-shot learning is an endeavour to transcend this capability down to machines. Inspired by the promise and power of probabilistic deep learning, we propose a novel variational inference network for few-shot classification (coined as `TRIDENT`) to decouple the representation of an image into *context* and *label* latent variables, and simultaneously infer them in an intertwined fashion. To induce *task-awareness*, as part of the inference mechanics of `TRIDENT`, we exploit information across both query and support images of a few-shot task using a novel built-in attention-based transductive feature extraction module (we call `AttFEX`). Our extensive experimental results corroborate the efficacy of `TRIDENT` and demonstrate that, using the simplest of backbones and a meta-learning strategy, it sets a new state-of-the-art in the most commonly adopted datasets *mini*ImageNet and *tiered*ImageNet (offering up to 4% and 5% improvements, respectively), as well as for the recent challenging cross-domain *mini*Imagenet → CUB scenario offering a significant margin (up to 20% improvement) beyond the best existing baselines[1].

## 1 Introduction

Deep learning algorithms are usually data hungry and require massive amounts of training data to reach a satisfactory level of performance on any task. To tackle this limitation, few-shot classification aims to learn to classify images from various unseen tasks in a data-deficient setting. In this exciting space, *metric learning* proposes to learn a shared feature extractor to embed the samples into a metric space of aggregated class embeddings (Sung et al., 2018; Vinyals et al., 2016; Snell et al., 2017; Wang et al., 2019; Liu et al., 2020). Due to limited data per class, these embeddings suffer from sample-bias and fail to efficiently represent class characteristics. Furthermore, sharing a feature extractor across tasks implies that the discriminative information learnt from the seen classes are equally effective on any arbitrary unseen classes, which is not true in most cases. *Transductive task-aware* few-shot learning approaches (Bateni et al., 2022; Ye et al., 2020; Cui & Guo, 2021) address these limitations by exploiting information hidden in the unlabeled data. As a result, the model learns task-specific embeddings by aligning the features of the labelled and unlabelled task instances for optimal distance metric based label assignment. Since the alignment of these embeddings is still subject to the relevance of the characteristics captured by the shared feature extractors, task-aware methods sometimes fail to extract meaningful representations particularly relevant to classification. *Probabilistic* methods address sample-bias by relaxing the need to find point estimates to approximate data-dependent distributions of either high-dimensional model weights (Nguyen et al., 2019; Ravi & Beatson, 2019; Gordon et al., 2019; Hu et al., 2020) or lower-dimensional class prototypes (Sun et al., 2021; Zhang et al., 2019). However, inferring a high-dimensional posterior of model parameters is inefficient in low-data regimes and estimating distributions of class prototypes involves using hand-crafted non-parametric aggregation techniques which may not be well suited for every unseen task.

---

[1]Codebase available at `https://github.com/anujinho/trident`.

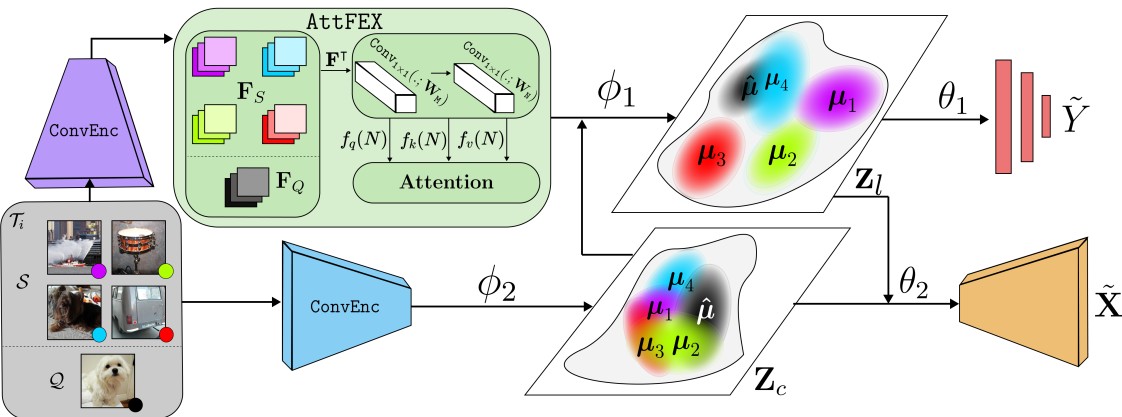

Figure 1: High-level process flow of `TRIDENT`. Inferred label latent variable $\mathbf{z}_l$ contains class-characterizing information, as is reflected by better separation of the distributions when compared to their context latent counterparts $\mathbf{z}_c$. `AttFEX` module generates *task-aware* feature maps by exploiting information from both support and query images, which compensates for the lack of label vectors $Y$ in inferring $\mathbf{z}_l$.

Although fit for purpose, all these approaches seem to overlook an important perspective. An image is composed of different attributes such as style, design, backdrop and setting which are not necessarily relevant discriminative characteristics for classification. Here, we refer to these attributes as *contextual* information. On the other hand, other class-characterizing attributes (such as wings of a bird, trunk of an elephant, hump on a camel's back) are critical for classification, irrespective of context. We refer to such attributes as *label* information. Typically, contextual information is majorly governed by context attributes, whereas the label characteristics are subtly embedded throughout an image. In other words, contextual information can be predominantly present across an image, whereas *attending* to subtle label information determines how effective a classification algorithm would be. Thus, we argue that attention to label-specific information should be ingrained into the mechanics of the classifier, decoupling it from contextual information. This becomes even more important in a few-shot setting where the network has to quickly learn from little data. Building upon this idea, we propose **tr**ansductive variational **i**nference of **de**coupled late**nt** variables (coined as `TRIDENT`), to simultaneously infer decoupled label and context information using two intertwined variational networks. To induce task-awareness while constructing the variational inference mechanics of `TRIDENT`, we introduce a novel **at**tention-based **t**ransductive **f**eature **ex**traction module (we call `AttFEX`) which further enhances the discriminative power of the inferred label attributes. This way `TRIDENT` infers distributions instead of point estimates and injects a handcrafted inductive-bias into the network to guide the classification process. Our main contributions can be summarized as:

1. We propose `TRIDENT`, a variational inference network to simultaneously infer two salient *decoupled* attributes of an image (*label* and *context*), by inferring these two using two intertwined variational sub-networks (Fig. 1).

2. We introduce an attention-based transductive feature extraction module, `AttFEX`, to enable `TRIDENT` see through and compare all images within a task, inducing transductive task-cognizance in the inference of label information.

3. We perform extensive evaluations to demonstrate that `TRIDENT` sets a new state-of-the-art by outperforming all existing baselines on the most commonly adopted datasets *mini*Imagenet and *tiered*Imagenet (up to 4% and 5%), as well as for the challenging cross-domain scenario of *mini*Imagenet $\rightarrow$ CUB (up to 20% improvement).

## 2 Related Work

**Metric-based learning.** This body of work involves mapping input samples into a lower-dimensional embedding space and then classifying the unlabelled samples based on a distance or similarity metric. By

parameterizing these mappings with neural networks and using differentiable similarity metrics for classification, these networks can be trained in an episodic manner (Vinyals et al., 2016) to perform few-shot classification. Prototypical Nets (Snell et al., 2017), Simple Shot (Wang et al., 2019), FRN (Wertheimer et al., 2021), Relation Networks (Sung et al., 2018), Matching Networks (Vinyals et al., 2016) variants of Graph Neural Nets (Satorras & Estrach, 2018; Yang et al., 2020), are a few examples of seminal ideas here.

**Transductive Feature-Extraction and Inference.** Transductive feature extraction or transductive task-aware learning is a variant of metric-learning with an adaptation mechanism that *aligns* support and query feature vectors in the embedding space for better representation of task-specific discriminative information. This not only improves the discriminative ability of classifiers across tasks, but also alleviates the problem of overfitting on limited support set since information from the query set is also used for extracting features of images in a task. CNAPS (Requeima et al., 2019), Transductive-CNAPS (Bateni et al., 2022), FEAT (Ye et al., 2020), Assoc-Align (Afrasiyabi et al., 2020), TPMN (Wu et al., 2021) and CTM (Li et al., 2019) are prime examples of such methods. Next to transduction for task-aware feature extraction, there are methods that use *transductive inference* to classify all the query samples at once by jointly assigning them labels, as opposed to their inductive counterparts where prediction is done on the samples one at a time. This is either done by iteratively propagating labels from the support to the query samples or by fine-tuning a pre-trained backbone using an additional entropy loss on all query samples, which encourages confident class predictions at query samples. TPN (Liu et al., 2019), Ent-Min (Dhillon et al., 2020), TIM (Boudiaf et al., 2020), Transductive-CNAPS (Bateni et al., 2022), LaplacianShot (Ziko et al., 2020), DPGN (Yang et al., 2020) and ReRank (SHEN et al., 2021) are a few notable examples in this space that usually report state-of-the-art results in certain few-shot classification settings (Liu et al., 2019). That being said, `TRIDENT` can be regarded as a transductive feature-extraction method, owing to `AttFEX`'s unique ability to see through and compare all images within a task.

**Optimization-based meta-learning.** These methods optimize for model parameters that are sensitive to task objective functions for fast gradient-based adaptation to new tasks. MAML (Finn et al., 2017) and its variants (Rajeswaran et al., 2019; Nichol et al., 2018b), (Oh et al., 2021) are a few prominent examples while LEO (Rusu et al., 2019) efficiently meta-updates its parameters in a lower dimensional latent space. Meta-learner LSTM (Ravi & Larochelle, 2017b) uses a separate meta-learner model to learn the exact optimization algorithm used to train another 'learner' neural network classifier.

**Probabilistic learning.** The estimated parameters of typical gradient-based meta-learning methods discussed earlier (Finn et al., 2017; Rusu et al., 2019; Mishra et al., 2018; Nichol et al., 2018b; Rajeswaran et al., 2019), have high variance due to the small task sample size. To deal with this, a natural extension is to model the uncertainty by treating these parameters as latent variables in a Bayesian framework as proposed in Neural Statistician (Edwards & Storkey, 2017), PLATIPUS (Finn et al., 2018), VAMPIRE (Nguyen et al., 2019), ABML (Ravi & Beatson, 2019), VERSA (Gordon et al., 2019), SIB (Hu et al., 2020), SAMOVAR (Iakovleva et al., 2020). Methods like ABPML (Sun et al., 2021) and VariationalFSL (Zhang et al., 2019) infer latent variables of class prototypes to perform classification and avoid inferring high-dimensional model parameters. ABPML (Sun et al., 2021) and VariationalFSL (Zhang et al., 2019) are the closest to our approach. In contrast to these two methods, we avoid hand-crafting class-level aggregations. Additionally, we enhance variational inference by incorporating a classification-relevant inductive bias through decoupling of label and context information.

## 3 Problem Definition

Consider a labelled dataset $\mathcal{D} = \{(\mathbf{x}_i, y_i) \,|\, i \in [1, N']\}$ of images $\mathbf{x}_i$ and class labels $y_i$. This dataset $\mathcal{D}$ is divided into three disjoint subsets: $\mathcal{D} = \{\mathcal{D}^{tr} \cup \mathcal{D}^{val} \cup \mathcal{D}^{test}\}$, respectively, referring to the training, validation, and test subsets. The validation dataset $\mathcal{D}^{val}$ is used for model selection and the testing dataset $\mathcal{D}^{test}$ for final evaluation. Following standard few-shot classification settings, as proposed in Vinyals et al. (2016); Sung et al. (2018); Snell et al. (2017), we use episodic training on a set of tasks $\mathcal{T}_i \sim p(\mathcal{T})$. The tasks are constructed by drawing $K$ random samples from $N$ different classes, which we denote as an ($N$-way, $K$-shot) task. Concretely, each task $\mathcal{T}_i$ is composed of a *support* and a *query* set. The support set $\mathcal{S} = \{(\mathbf{x}_{kn}^S, y_{kn}^S) \,|\, k \in [1, K], n \in [1, N]\}$ contains $K$ samples per class and the query set $\mathcal{Q} = \{(\mathbf{x}_{kn}^Q, y_{kn}^Q) \,|\, k \in$

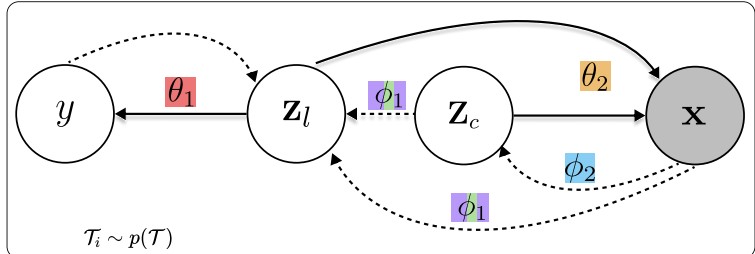

Figure 2: Generative Model of TRIDENT. Dotted lines indicate variational inference and solid lines refer to generative processes. The inference and generative parameters are color coded to correspond to their respective architectures indicated in Fig.1 and Fig.4.

$[1, Q], n \in [1, N]\}$ contains $Q$ samples per class. For a given task, the $NQ$ query and $NK$ support images are disjoint to assess the generalization performance.

## 4 The Proposed Method: TRIDENT

Let us start with the high-level idea. The proposed approach is devised to learn meaningful representations that capture two pivotal characteristics of an image by modelling them as separate latent variables: (i) $\mathbf{z}_c$ representing *context*, and (ii) $\mathbf{z}_l$ embodying class *labels*. Inferring these two latent variables simultaneously allows $\mathbf{z}_l$ to learn meaningful distributions of class-discriminating characteristics *decoupled* from context features represented by $\mathbf{z}_c$. We argue that learning $\mathbf{z}_l$ as the sole latent variable for classification results in capturing a mixture of true label and other context information. This in turn can lead to sub-optimal classification performance, especially in a few-shot setting where the information per class is scarce and the network has to adapt and generalize quickly. By inferring decoupled label and context latent variables, we inject a handcrafted inductive-bias that incorporates only relevant characteristics, and thus, ameliorates the network's classification performance.

### 4.1 Generative Process

The directed graphical model in Fig. 2 illustrates the common underlying generative process $p$ such that $p_i = p(\mathbf{x}_i, y_i \mid \mathbf{z}_{li}, \mathbf{z}_{ci})$. For the sake of brevity, in the following we drop the sample index $i$ as we always refer to terms associated with a single data sample. We work on the logical premise that the label latent variable $\mathbf{z}_l$ is responsible for generating class label as well as for image reconstruction, whereas the context latent variable $\mathbf{z}_c$ is only responsible for image reconstruction (solid lines in the figure). Formally, the data is explained by the generative processes: $p_{\theta_1}(y \mid \mathbf{z}_l) = \text{Cat}(y \mid \mathbf{z}_l)$ and $p_{\theta_2}(\mathbf{x} \mid \mathbf{z}_l, \mathbf{z}_c) = g_{\theta_2}(\mathbf{x}; \mathbf{z}_l, \mathbf{z}_c)$, where $\text{Cat}(.)$ refers to a multinomial distribution and $g_{\theta_2}(\mathbf{x}; \mathbf{z}_l, \mathbf{z}_c)$ is a suitable likelihood function such as a Gaussian or Bernoulli distribution. The likelihoods of both these generative processes are parameterized using deep neural networks and the priors of the latent variables are chosen to be standard multivariate Gaussian distributions (Kingma & Welling, 2014; Kingma et al., 2014): $p(\mathbf{z}_c) = \mathcal{N}(\mathbf{z}_c \mid \mathbf{0}, \mathbf{I})$ and $p(\mathbf{z}_l) = \mathcal{N}(\mathbf{z}_l \mid \mathbf{0}, \mathbf{I})$.

### 4.2 Variational Inference of Decoupled $\mathbf{Z}_l$ and $\mathbf{Z}_c$

Computing exact posterior distributions is intractable due to high dimensionality and non-linearity of the deep neural network parameter space. Following Kingma & Welling (2014); Kingma et al. (2014), we instead construct an approximate posterior over the latent variables by introducing a fixed-form distribution $q(\mathbf{z}_l, \mathbf{z}_c \mid \mathbf{x}, y)$ parameterized by $\phi$. By using $q_\phi(.)$ as an inference network, the inference is rendered tractable, scalable and amortized since $\phi$ now acts as the global variational parameter. We assume $q_\phi$ has a factorized form $q_\phi(\mathbf{z}_c, \mathbf{z}_l \mid \mathbf{x}, y) = q_{\phi_1}(\mathbf{z}_l \mid \mathbf{x}, \mathbf{z}_c) q_{\phi_2}(\mathbf{z}_c \mid \mathbf{x})$, where $q_{\phi_1}(.), q_{\phi_2}(.)$ are assumed to be multivariate Gaussian distributions. As is also depicted in Fig. 2, we use $\mathbf{z}_c$ as input to $q_{\phi_1}(.)$ to infer $\mathbf{z}_l$ because of their conditional dependence given $\mathbf{x}$. This way we forge a path to allow *necessary* context latent information flow through the label inference network. On the other hand, the opposite direction (using $\mathbf{z}_l$ to infer $\mathbf{z}_c$) is unnecessary,

because label information does not directly contribute to the extraction of context features. We will further reflect on this design choice in the next subsection. Neural networks are then used to parameterize both inference networks as:

$$
\begin{aligned}
q_{\phi_2}\left(\mathbf{z}_c \,|\, \mathbf{x}\right) &= \mathcal{N}\left(\mathbf{z}_c \,|\, \boldsymbol{\mu}_{\phi_2}(\mathbf{x}), diag(\boldsymbol{\sigma}^2_{\phi_2}(\mathbf{x}))\right), \\
q_{\phi_1}\left(\mathbf{z}_l \,|\, \mathbf{x}, \mathbf{z}_c\right) &= \mathcal{N}\left(\mathbf{z}_l \,|\, \boldsymbol{\mu}_{\phi_1}(\mathbf{x}, \mathbf{z}_c), diag(\boldsymbol{\sigma}^2_{\phi_1}(\mathbf{x}, \mathbf{z}_c))\right).
\end{aligned}
\tag{1}
$$

To find the optimal *approximate* posterior, we derive the evidence lower bound (ELBO) on the marginal likelihood of the data to form our objective function:

$$
\begin{aligned}
p(\mathbf{x}, y) &= \iint p(\mathbf{x}, y \,|\, \mathbf{z}_c, \mathbf{z}_l)\, p(\mathbf{z}_s, \mathbf{z}_l)\, d\mathbf{z}_c\, d\mathbf{z}_l, \\
&= \mathbb{E}_{q(\mathbf{z}_c, \mathbf{z}_l \,|\, x)}\left[\frac{p(\mathbf{x} \,|\, \mathbf{z}_l, \mathbf{z}_c)p(y \,|\, \mathbf{z}_l)p(\mathbf{z}_l)p(\mathbf{z}_c)}{q(\mathbf{z}_l, \mathbf{z}_c \,|\, \mathbf{x})}\right]. \\
\ln p(\mathbf{x}, y) &\geqslant \mathbb{E}_{q(\mathbf{z}_c, \mathbf{z}_l \,|\, \mathbf{x})}\left[\ln\left(\frac{p(\mathbf{x} \,|\, \mathbf{z}_l, \mathbf{z}_c)p(y \,|\, \mathbf{z}_l)p(\mathbf{z}_l)p(\mathbf{z}_c)}{q(\mathbf{z}_c, \mathbf{z}_l \,|\, \mathbf{x})}\right)\right], \\
&= \mathbb{E}_{q_{\phi_2}}\left[\mathbb{E}_{q_{\phi_1}}\left[\ln\left(\frac{p(\mathbf{x} \,|\, \mathbf{z}_c, \mathbf{z}_l)p(y \,|\, \mathbf{z}_l)p(\mathbf{z}_c)p(\mathbf{z}_l)}{q(\mathbf{z}_c \,|\, \mathbf{x})q(\mathbf{z}_l \,|\, \mathbf{x}, \mathbf{z}_c)}\right)\right]\right].
\end{aligned}
$$

Denoting $\Psi = (\theta_1, \theta_2, \phi_1, \phi_2)$, the negative ELBO can be given by

$$
\begin{aligned}
\mathcal{L}(\Psi) = &-\mathbb{E}_{q_{\phi_2}}\mathbb{E}_{q_{\phi_1}}\left[\ln p_{\theta_2}(\mathbf{x} \,|\, \mathbf{z}_c, \mathbf{z}_l) + \ln p_{\theta_1}(y \,|\, \mathbf{z}_l)\right] + \\
&\mathbb{E}_{q_{\phi_2}}\left[D_{KL}\big(q_{\phi_1}(\mathbf{z}_l \,|\, \mathbf{x}, \mathbf{z}_c)\,\|\, p(\mathbf{z}_l)\big)\right] + \\
&D_{KL}\big(q_{\phi_2}(\mathbf{z}_c \,|\, \mathbf{x})\,\|\, p(\mathbf{z}_c)\big),
\end{aligned}
\tag{2}
$$

where the second line follows the graphical model in Fig 2, and $\mathbb{E}(.)$ and $\ln(.)$ denote the expectation operator and the natural logarithm, respectively. We avoid computing biased gradients by following the re-parameterization trick from Kingma & Welling (2014). Note that in equation 1 we deliberately choose to exclude the label information $y$ as input to $q_{\phi_1}(.)$ to be able to exploit the associated generative network $p_{\theta_1}(y \,|\, \mathbf{z}_l)$ as a classifier. The consequence and the proposed solution to accommodate this design choice are discussed in the next subsection.

### 4.3 `AttFEX` for Transductive Feature Extraction

Our design choice to omit label information $y$ when inferring $\mathbf{z}_l$ (as discussed for equation 1) can be an information bottleneck and counter-productive to the discriminative power $\mathbf{z}_l$ holds. However, this allows us to employ $\mathbf{z}_l$ for classification and not reconstruction of the label. To compensate for this bottleneck, we introduce an attention-based transductive feature extractor (`AttFEX`) module that allows the network $q_{\phi_1}(\mathbf{z}_l \,|\, \mathbf{x}, \mathbf{z}_c)$ see through and compare images across all classes within each task (irrespective of being from the query or support sets), thus, induces *task-cognizance* in the inference network. We first extract the feature maps of all images in the task using a convolutional block $\mathbf{F} = \texttt{ConvEnc}(\mathbf{X})$ where $\mathbf{X} \in \mathbb{R}^{N(K+Q) \times C \times W \times H}$, $\mathbf{F} \in \mathbb{R}^{N(K+Q) \times C' \times W' \times H'}$. The feature map tensor $\mathbf{F}$ is then transposed into $\mathbf{F}' \in \mathbb{R}^{C' \times N(K+Q) \times W' \times H'}$ and fed into two consecutive $1 \times 1$ convolution blocks. This helps the network utilize information across corresponding pixels of all images in a task $\mathcal{T}_i$, which can be considered as a parametric comparison of classes. We leverage the fact that `ConvEnc` already extracts local pixel information by using larger kernels, and thus, use parameter-light $1 \times 1$ convolutions subsequently to focus only on individual pixels. Let $\mathbf{F}'_i$ denote the $i^{th}$ channel (or feature map layer) out of total of $C'$ available and `ReLU` denote the rectified linear unit activation. The $1 \times 1$ convolution block ($\texttt{Conv}_{1 \times 1}$) is formulated as follows:

$$
\begin{aligned}
\mathbf{M}_i &= \texttt{ReLU}\big(\texttt{Conv}_{1 \times 1}(\mathbf{F}'_i, \mathbf{W}_M)\big), \forall i \in [1, C']; \\
\mathbf{N}_j &= \texttt{ReLU}\big(\texttt{Conv}_{1 \times 1}(\mathbf{M}_j, \mathbf{W}_N)\big), \forall j \in [1, C'];
\end{aligned}
\tag{3}
$$

where $\mathbf{N} \in \mathbb{R}^{C' \times 32 \times W' \times H'}$ and $\mathbf{W}_M \in \mathbb{R}^{64 \times N(K+Q) \times 1 \times 1}$, $\mathbf{W}_N \in \mathbb{R}^{32 \times 64 \times 1 \times 1}$ denote the learnable weights. Next, we want to blend information across feature maps for which we use a self-attention mechanism (Vaswani

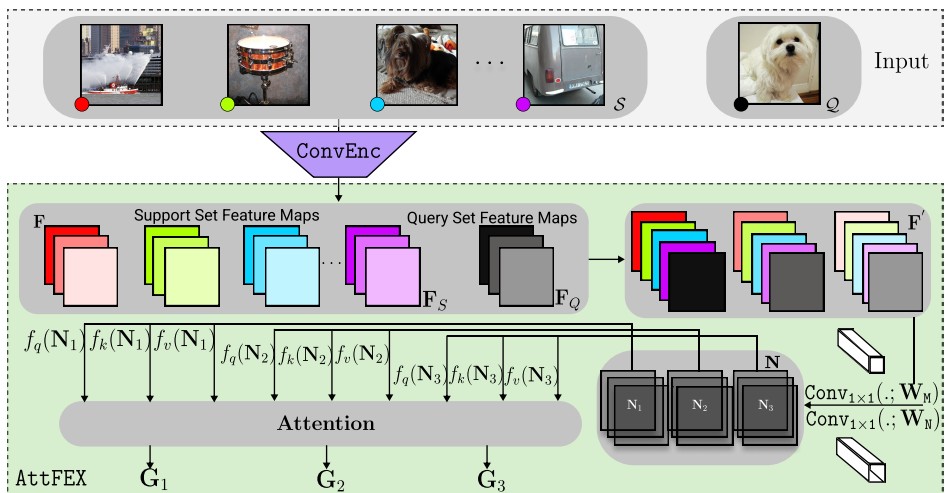

Figure 3: `AttFEX` module depicting colors as images and shades as feature maps. We illustrate only 3 image feature maps and 3 channels instead of 32 for $\mathbf{N}$, for the sake of simplicity.

et al., 2017) across $\mathbf{N}_j, \forall j \in [1, 32]$. To do so, we feed $\mathbf{N}$ to query, key and value extraction networks $f_q(,;\mathbf{W}_Q)$, $f_k(.;\mathbf{W}_K)$, $f_v(.;\mathbf{W}_V)$ which are also designed to be $1 \times 1$ convolutions as:

$$
\begin{aligned}
\mathbf{Q}_i &= \texttt{ReLU}\left(\texttt{Conv}_{1\times1}(\mathbf{N}_i, \mathbf{W}_Q)\right), \quad \forall i \in [1, C']; \\
\mathbf{K}_i &= \texttt{ReLU}\left(\texttt{Conv}_{1\times1}(\mathbf{N}_i, \mathbf{W}_K)\right), \quad \forall i \in [1, C']; \\
\mathbf{V}_i &= \texttt{ReLU}\left(\texttt{Conv}_{1\times1}(\mathbf{N}_i, \mathbf{W}_V)\right), \quad \forall i \in [1, C'];
\end{aligned}
\tag{4}
$$

where $\mathbf{W}_Q, \mathbf{W}_K, \mathbf{W}_V \in \mathbb{R}^{1\times32\times1\times1}$ are the learnable weights and $\mathbf{Q}, \mathbf{K}, \mathbf{V} \in \mathbb{R}^{C'\times1\times W'\times H'}$ are the query, key and value tensors. Next, each feature map $\mathbf{N}_j$ is mapped to its output tensor $\mathbf{G}_j$ by computing a weighted sum of the values, where each weight (within parentheses in equation 5) measures the compatibility (or similarity) between the query and its corresponding key tensor using an inner-product:

$$
\mathbf{G}_i = \sum_{j=1}^{C'} \left( \frac{\exp\left(\mathbf{Q}_i \cdot \mathbf{K}_j\right)}{\sqrt{d_k} \cdot \sum_{k=1}^{C'} \exp\left(\mathbf{Q}_i \cdot \mathbf{K}_k\right)} \right) \mathbf{V}_i,
\tag{5}
$$

where $d_k = W' \times H'$, and $\mathbf{G}_i \in \mathbb{R}^{1\times C'\times W'\times H'}$, $\forall i$. Finally, we transform the original feature maps $\mathbf{F}$ by applying a Hadamard product between the feature mask $\mathbf{G}$ and $\mathbf{F}$, thus, rendering the required feature maps transductive:

$$
\tilde{\mathbf{F}}^S = \mathbf{G} \circ \mathbf{F}^S \quad or \quad \tilde{\mathbf{F}}^Q = \mathbf{G} \circ \mathbf{F}^Q.
$$

Here, $\mathbf{F}^S$ and $\mathbf{F}^Q$ represent the feature maps corresponding to the support and query images, respectively. As a result of operating on this channel-pixel distribution across images in a task, $\tilde{\mathbf{F}}^S$ and $\tilde{\mathbf{F}}^Q$ are rendered transductive. Unlike other attention-based few-shot learning methods (Ye et al., 2020; Vinyals et al., 2016), we do not compute an attention-based transform on the flattened support and query vectors, but rather on the outputs of the $\texttt{Conv}_{1\times1}(.;\mathbf{W}_N)$ to effectively fuse information from multiple class-pixel comparisons. Note that the query tensor $\mathbf{Q}$ must not be confused with the query set $\mathcal{Q}$ of a task.

### 4.4 `TRIDENT`'s Transductive ELBO

`AttFEX`'s transductive feature extraction process introduces task-level dependencies in the variational formulation of $q_{\phi_1}$. To incorporate this dependency in equation 2, we now revise the derivation of our negative ELBO to be defined in terms of the entire task set and not individual data points. Let $\mathbf{X} = \mathbf{X}^{\mathcal{S}} \cup \mathbf{X}^{\mathcal{Q}}$ denote the tensor containing all images sampled in a task, $Y = Y^{\mathcal{S}} \cup Y^{\mathcal{Q}}$ denote all the labels corresponding to the

images in the task and $N' = NK + NQ$ be the total number of samples in a task. Considering all samples to be independently and identically distributed (I.I.D.), the likelihood of the entire task can be written as:

$$p(\mathbf{X}, Y) = \prod_{i=1}^{N'} \iint p(\mathbf{x}_i, y_i \mid \mathbf{z}_{ci}, \mathbf{z}_{li}) \, p(\mathbf{z}_{ci}, \mathbf{z}_{li}) \, d\mathbf{z}_{ci} \, d\mathbf{z}_{li}. \tag{6}$$

Since the generative networks $p_{\theta_2}(\mathbf{x} \mid \mathbf{z}_c, \mathbf{z}_l)$ and $p_{\theta_1}(y \mid \mathbf{z}_l)$ remain inductive, while the approximate inference network $q_{\phi_1}(\mathbf{z}_l \mid \mathbf{X}, \mathbf{z}_c)$ becomes transductive (via `AttFEX`), the log-likelihood now becomes:

$$\ln p(\mathbf{X}, Y) \geq \sum_{i=1}^{N'} \mathbb{E}_{q_{\phi_2}} \left[ \mathbb{E}_{q_{\phi_1}} \left[ \ln \left( \frac{p(\mathbf{x}_i \mid \mathbf{z}_{ci}, \mathbf{z}_{li}) p(y \mid \mathbf{z}_{li}) p(\mathbf{z}_c) p(\mathbf{z}_l)}{q(\mathbf{z}_{ci} \mid \mathbf{x}_i) q(\mathbf{z}_{li} \mid \mathbf{X}, \mathbf{z}_{ci})} \right) \right] \right]. \tag{7}$$

Finally, the overall negative ELBO for the entire task can be given by

$$\begin{aligned}
\mathcal{L}(\Psi) = & -\sum_{i=1}^{N'} \mathbb{E}_{q_{\phi_2}} \mathbb{E}_{q_{\phi_1}} \left[ \ln p_{\theta_2}(\mathbf{x}_i \mid \mathbf{z}_{ci}, \mathbf{z}_{li}) + \ln p_{\theta_1}(y_i \mid \mathbf{z}_{li}) \right] + \\
& \mathbb{E}_{q_{\phi_2}} \left[ D_{KL}\big( q_{\phi_1}(\mathbf{z}_{li} \mid \mathbf{X}, \mathbf{z}_{ci}) \| \, p(\mathbf{z}_l) \big) \right] + \\
& D_{KL}\big( q_{\phi_2}(\mathbf{z}_{ci} \mid \mathbf{x}_i) \| \, p(\mathbf{z}_c) \big).
\end{aligned} \tag{8}$$

Assuming Gaussian distributions for the priors as well as the variational distributions allows us to compute the KL Divergences of $\mathbf{z}_l$ and $\mathbf{z}_c$ (last two terms in equation 8) analytically (Kingma & Welling, 2014). By considering a multivariate Gaussian distribution and a multinomial distribution as the likelihood functions for $p_{\theta_2}(\mathbf{x} \mid \mathbf{z}_c, \mathbf{z}_l)$ and $p_{\theta_1}(y \mid \mathbf{z}_l)$, respectively, the negative log-likelihood of $\mathbf{x}$ becomes the mean squared error (MSE) between the reconstructed images $\tilde{\mathbf{x}}$ and the ground-truth images $\mathbf{x}$ while the negative log-likelihood of $y$ becomes the cross-entropy between the actual labels $y$ and the predicted labels $\tilde{y}$. After working equation 8 out, we arrive at our overall objective function $\mathcal{L} = \mathcal{L}_R + \mathcal{L}_C$, where:

$$\begin{aligned}
\mathcal{L}_R &= \alpha_1 \sum_{i=1}^{N'} \|\mathbf{x}_i - \tilde{\mathbf{x}}_i\|^2 - KL(\mu_{ci}, \sigma_{ci}), \\
\mathcal{L}_C &= -\alpha_2 \sum_{i=1}^{N'} \sum_{n=1}^{N} [y_i]_n \ln p_{\theta_1}(\tilde{y}_i = n \mid \mathbf{z}_l) - KL(\mu_{li}, \sigma_{li}).
\end{aligned} \tag{9}$$

where $KL(\mu, \sigma) = \frac{1}{2} \sum_{d=1}^{D} \left( 1 + 2\ln(\sigma^d) - (\mu^d)^2 - (\sigma^d)^2 \right)$, $[y_i]_n$ denotes the $n$-th dimension of the $i$-th one-hot encoded ground-truth vector $y$, $D$ denotes the dimension of the latent space, $N$ is the total number of classes in an ($N$-way, $K$-shot) task, $\alpha_1, \alpha_2$ are constant scaling factors, $\mu_c$ and $\sigma_c^2$ denote the mean and variance vectors of context latent distribution, and $\mu_l$ and $\sigma_l^2$ denote the mean and variance vectors of label latent distribution. The hyper-parameters $\alpha_1, \alpha_2$ only scale the evidence lower-bound appropriately, since the reconstruction loss is in practice three orders of magnitude greater than the cross-entropy loss. Moreover, these scaling factors can be understood as gradient-scaling parameters which help improve training in heterogeneous likelihoods (Gaussian and Categorical in our case) (Javaloy et al., 2022).

## 4.5 Algorithmic Overview and Training Strategy

**Overview of `TRIDENT`.** The complete architecture of `TRIDENT` is illustrated in Fig. 4. The `ConvEnc` feature extractor and the linear layers $\mu_{\phi_2}(.)$, $\sigma_{\phi_2}^2(.)$ constitute the inference network $q_{\phi_2}$ of the context latent variable (bottom row of Fig. 4). The `AttFEX` module, another `ConvEnc`, and linear layers $\mu_{\phi_1}(.)$ and $\sigma_{\phi_1}^2(.)$ make up the inference network $q_{\phi_1}$ of the label latent variable (top row of Fig. 4). The proposed approach, `TRIDENT`, is described in Algorithm 1. Note that `TRIDENT` is trained in a MAML (Finn et al., 2017) fashion, where depending on the inner or outer loop, the support or query set ($g \in \{\mathcal{S}, \mathcal{Q}\}$) will be the reference, respectively. First, the lower `ConvEnc` block extracts feature maps $\mathbf{X}_{\text{CE}}^g = \text{ConvEnc}(\mathbf{X}^g)$. $\mathbf{X}_{\text{CE}}^g$'s are then flattened and passed onto $\mu_{\phi_2}(.)$, $\sigma_{\phi_2}^2(.)$, which respectively output the mean and variance vectors of the

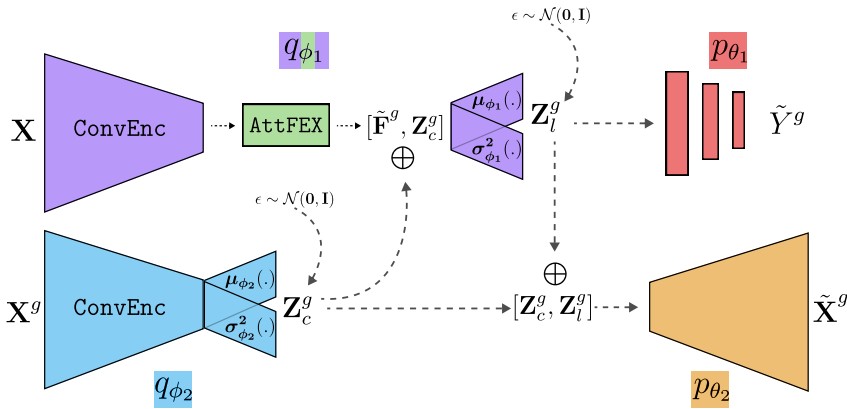

Figure 4: TRIDENT is comprised of two intertwined variational networks. $\mathbf{Z}_c{}^g$ is concatenated with the output of AttFEX, and used for inferring $\mathbf{Z}_l^g$, where $g \in \{\mathcal{S}, \mathcal{Q}\}$. Next, both $\mathbf{Z}_l^g$ and $\mathbf{Z}_c{}^g$ are used to reconstruct images $\tilde{\mathbf{X}}^g$ while $\mathbf{Z}_l^g$ is used to extract $\tilde{Y}^g$.

---

**Algorithm 1:** TRIDENT

---

**Require:** $\mathbf{X}^{\mathcal{S}}, \mathbf{X}^{\mathcal{Q}}, Y^g, \mathbf{X}_{\text{CE}}^g$, where $g \in \{\mathcal{S}, \mathcal{Q}\}$

1 Sample: $\mathbf{Z}_c{}^g \sim q_{\phi_2}(\mathbf{Z}_c \,|\, \boldsymbol{\mu}_{\phi_2}(\mathbf{X}_{\text{CE}}^g),\, diag(\boldsymbol{\sigma}_{\phi_2}^2(\mathbf{X}_{\text{CE}}^g)))$

2 Compute *task-cognizant* embeddings: $[\tilde{\mathbf{F}}^{\mathcal{S}}, \tilde{\mathbf{F}}^{\mathcal{Q}}] = \text{AttFEX}(\text{ConvEnc}(\mathbf{X})); \mathbf{X} = \mathbf{X}^{\mathcal{S}} \cup \mathbf{X}^{\mathcal{Q}}$

3 Concatenate $\mathbf{Z}_c{}^g$ and $\tilde{\mathbf{F}}^g$ into $[\tilde{\mathbf{F}}^g, \mathbf{Z}_c{}^g]$ and sample: $\mathbf{Z}_l^g \sim q_{\phi_1}(\mathbf{Z}_l \,|\, \boldsymbol{\mu}_{\phi_1}([\tilde{\mathbf{F}}^g, \mathbf{Z}_c{}^g]),\, diag(\boldsymbol{\sigma}_{\phi_1}^2([\tilde{\mathbf{F}}^g, \mathbf{Z}_c{}^g])))$

4 Reconstruct $\mathbf{X}^g$ using $\tilde{\mathbf{X}}^g = p_{\theta_2}(\mathbf{X} \,|\, \mathbf{Z}_l^g, \mathbf{Z}_c{}^g)$

5 Extract class-conditional probabilities using: $p(\tilde{Y}^g \,|\, \mathbf{Z}_l^g) = \text{softmax}(p_{\theta_1}(Y^g \,|\, \mathbf{Z}_l^g))$

6 Compute $\mathcal{L}^g = \mathcal{L}_R^g + \mathcal{L}_C^g$ using equation 9

**Return:** $\mathcal{L}^g$

---

*context* latent distribution, as discussed in equation 1. This is done either for the entire support or the query images $\mathbf{X}^g$, where $g \in \{S, Q\}$ for a given task $\mathcal{T}_i$. We then sample a set of vectors $\mathbf{Z}_c^g$ (subscript $c$ for *context*) from their corresponding Gaussian distributions using the re-parameterization trick (line 1, Algorithm 1). Upon passing $\mathbf{X} = \mathbf{X}^{\mathcal{S}} \cup \mathbf{X}^{\mathcal{Q}}$ through the upper ConvEnc, the AttFEX module of $q_{\phi_1}$ comes into play to create *task-cognizant* feature maps $\tilde{\mathbf{F}}^g$ for either $\mathcal{S}$ or $\mathcal{Q}$ (line 2). $\mathbf{Z}_c^g$ together with $\tilde{\mathbf{F}}^g$ are passed onto the linear layers $\mu_{\phi_1}(.), \sigma_{\phi_1}^2(.)$ to generate the mean and variance vectors of the *label* latent Gaussian distributions (line 3). After sampling the set of vectors $\mathbf{Z}_l^g$ (subscript $l$ for *label*) from their corresponding distributions, we use $\mathbf{Z}_l^g$ and $\mathbf{Z}_c^g$ to reconstruct images $\tilde{\mathbf{X}}^g$ using the generative network $p_{\theta_2}$ (line 4). Next, $\mathbf{Z}_l^g$'s are input to the classifier network $p_{\theta_1}$ to generate the class logits, which are normalized using a softmax(.), resulting in class-conditional probabilities $p(\tilde{Y}^g \,|\, \mathbf{Z}_l^g)$ (line 5). Finally (in line 6), using the outputs of all the components discussed earlier, we calculate the loss $\mathcal{L}^g$ as formulated in equation 8, 9.

**Training strategy.** An important aspect of the training procedure of TRIDENT is that its set of parameters $\Psi = (\theta_1, \theta_2, \phi_1, \phi_2)$ are meta-learnt by back-propagating through the adaptation procedure on the support set, as proposed in MAML (Finn et al., 2017) and illustrated here in Algorithm 2. This increases the sensitivity of the parameters $\Psi$ towards the loss function for fast adaptation to unseen tasks and reduces generalization errors on the query set $\mathcal{Q}$, as discussed from a dynamical systems standpoint in Finn et al. (2017). First, we randomly initialize the parameters $\Psi$ (line 1, Algorithm 2) to compute the objective function over the support set $\mathcal{L}^{\mathcal{S}_i}(\Psi)$ using equation 9, and perform a number of gradient descent steps on the parameters $\Psi$ to adapt them to the support set (lines 5 to 9). This is called the *inner-update* and is done separately for all the support sets corresponding to their $B$ different tasks (line 3). Once the inner-update is computed for each of the $B$ parameter sets, the loss is evaluated on the query set $\mathcal{L}^{\mathcal{Q}_i}(\Psi_i')$ (line 12), following

---

**Algorithm 2:** End to End Meta-Training of `TRIDENT`

---

**Require:** $\mathcal{D}^{tr}$, $\alpha$, $\beta$, $B$

1   Randomly initialise $\Psi = (\phi_1, \phi_2, \theta_1, \theta_2)$

2   **while** *not converged* **do**

3      Sample $B$ tasks $\mathcal{T}_i = \mathcal{S}_i \cup \mathcal{Q}_i$ from $\mathcal{D}^{tr}$

4      **for** *each task $\mathcal{T}_i$* **do**

5         **for** *number of adaptation steps* **do**

6            Compute $\mathcal{L}^{\mathcal{S}_i}(\Psi) = \texttt{TRIDENT}(\mathcal{T}_i - \{Y^{\mathcal{Q}_i}\})$

7            Evaluate $\nabla_{(\Psi)}\mathcal{L}^{\mathcal{S}_i}(\Psi)$

8            $\Psi \leftarrow \Psi - \alpha\nabla_\Psi\mathcal{L}^{\mathcal{S}_i}(\Psi)$

9         **end**

10         $(\Psi')_i = \Psi$

11      **end**

12      Compute $\mathcal{L}^{\mathcal{Q}_i}(\Psi'_i) = \texttt{TRIDENT}(\mathcal{T}_i - \{Y^{\mathcal{S}_i}\}); \forall i \in [1, B]$

13      Meta-update on $\mathcal{Q}_i$: $\Psi \leftarrow \Psi - \beta\nabla_\Psi \sum_{i=1}^{B} \mathcal{L}^{\mathcal{Q}_i}(\Psi'_i)$

14   **end**

---

which a *meta-update* is conducted over all the corresponding query sets, which involves computing a gradient through a gradient procedure as described in Finn et al. (2017) (line 13).

## 5   Experimental Evaluation

The goal of this section is to address the following four questions: (i) How well does `TRIDENT` perform when compared against the state-of-the-art methods for few-shot classification? (ii) How reliable is `TRIDENT` in terms of the confidence and uncertainty metrics? (iii) How well does `TRIDENT` perform in a cross-domain setting where there is a domain shift between the training and testing datasets? (iv) Does `TRIDENT` actually decouple latent variables?

**Benchmark Datasets.** We evaluate `TRIDENT` on the three most commonly adopted datasets: *mini*Imagenet (Ravi & Larochelle, 2017a), *tiered*Imagenet (Ren et al., 2018) and CUB (Welinder et al., 2010). ***mini*Imagenet** (Vinyals et al., 2016) is a subset of ImageNet (Deng et al., 2009) for few-shot classification. It contains 100 classes with 600 samples each. We follow the predominantly adopted settings of Ravi & Larochelle (2017a); Chen et al. (2019) where we split the entire dataset into 64 classes for training, 16 for validation and 20 for testing. ***tiered*Imagenet** is a larger subset of ImageNet with 608 classes and $779,165$ total images, which are grouped into 34 higher-level nodes in the *ImageNet* human-curated hierarchy. This set of nodes is partitioned into 20, 6, and 8 disjoint sets of training, validation, and testing nodes, and the corresponding classes form the respective meta-sets. **CUB** (Welinder et al., 2010) dataset has a total of 200 classes, split into training, validation and test sets following Chen et al. (2019). We use this dataset to simulate the effect of a domain shift where the model is first trained on a (5-way, 1 or 5-shot) configuration of *mini*Imagenet and then tested on the test classes of CUB, as used in Chen et al. (2019); Boudiaf et al. (2020); Ziko et al. (2020); Long et al. (2018).

**Implementational Details.** We use PyTorch (Paszke et al., 2019) and learn2learn (Arnold et al., 2020) for all our implementations. We use a commonly adopted `Conv4` architecture (Ravi & Larochelle, 2017a; Finn et al., 2017; Patacchiola et al., 2020; Afrasiyabi et al., 2020; Wang et al., 2019; Boudiaf et al., 2020) as `ConvEnc` to obtain the generic feature maps. Following the standard setting in the literature (Finn et al., 2017; Ravi & Larochelle, 2017a), the `Conv4` has four convolutional blocks where each block has a $3 \times 3$ convolution layer with 32 feature maps, followed by a batch normalization (BN) (Ioffe & Szegedy, 2015) layer, a $2 \times 2$ max-pooling layer and a `LeakyReLU(0.2)` activation. The generative network $p_{\theta_1}$ for $\mathbf{z}_l$ is a classifier with two linear layers and a `LeakyReLU(0.2)` activation in between, while $p_{\theta_2}$ for $\mathbf{z}_c$ consists of four blocks of a 2-D upsampling layer, followed by a $3 \times 3$ convolution and `LeakyReLU(0.2)` activation. Both latent variables $\mathbf{z}_l$ and $\mathbf{z}_c$ have a dimensionality of 64.

Following Nichol et al. (2018a); Liu et al. (2019); Vaswani et al. (2017), images are resized to $84 \times 84$ for all configurations and we train and report test accuracy of (5-way, 1 and 5-shot) settings with 10 query images per class for all datasets. The hyperparameter values (**H.P.**) used for training TRIDENT on *mini*Imagenet and *tiered*Imagenet are shown in Table 1. We apply the same hyperparameters for the cross-domain testing scenario of *mini*Imagenet $\rightarrow$ CUB used for training TRIDENT on *mini*Imagenet,

Table 1: **H.P.** values when training TRIDENT.

| H.P. | *mini*Imagenet | | *tiered*Imagenet | |
|---|---|---|---|---|
| | 5-way, 1-shot | 5-way, 5-shot | 5-way, 1-shot | 5-way, 5-shot |
| $\alpha_1$ | 1e-2 | 1e-2 | 1e-2 | 1e-2 |
| $\alpha_2$ | 100 | 100 | 150 | 150 |
| $\alpha$ | 1e-3 | 1e-3 | 1.5e-3 | 1.7e-3 |
| $\beta$ | 1e-4 | 1e-4 | 1.5e-4 | 1.7e-4 |
| $B$ | 20 | 20 | 20 | 20 |
| $n$ | 5 | 5 | 5 | 5 |

for the given ($N$-way, $K$-shot) configuration. Hyperparameters are kept fixed throughout training, validation and testing for a given configuration. Adam (Kingma & Ba, 2015) optimizer is used for inner and meta-updates. Finally, the query, key and value extraction networks $f_q(,.; \mathbf{W}_Q)$, $f_k(.; \mathbf{W}_K)$, $f_v(.; \mathbf{W}_V)$ of the AttFEX module only use $\text{Conv}_{1 \times 1}(.)$ and not the LeakyReLU(0.2) activation function for (5-way, 1-shot) tasks, irrespective of the dataset. We observed that utilizing BatchNorm (Ioffe & Szegedy, 2015) in the decoder of $z_c$ ($p_{\theta_2}$) to train TRIDENT on (5-way, 5-shot) tasks of *mini*Imagenet and on (5-way, 1-shot) tasks of *tiered*Imagenet leads to better scores and improved stability during training. We used the ReLU activation function instead of LeakyReLU(0.2) to carry out training on (5-way, 1-shot) tasks of *tiered*Imagenet. Meta-learning objectives can lead to unstable optimization processes in practice, especially when coupled with stochastic sampling in latent spaces, as also previously observed in Antreas Antoniou et al. (2019); Rusu et al. (2019). For ease of experimentation, we clip the meta-gradient norm at an absolute value of 1. Since AttFEX operates on all samples available in a task, scaling to a larger number of ways and shots per task requires more computational resources. TRIDENT converges in $82,000$ and $22,500$ epochs for (5-way, 1-shot) and (5-way, 5-shot) tasks of *mini*Imagenet, respectively and takes $67,500$ and $48,000$ epochs for convergence on (5-way, 1-shot) and (5-way, 5-shot) tasks of *tiered*Imagenet, respectively. This translates to an average training time of 110 hours on an 11GB NVIDIA 1080Ti GPU. Note that we did not employ any data augmentation, feature averaging or any other data apart from the corresponding training subset $\mathcal{D}^{tr}$, during training.

## 5.1 Evaluation Results

We report test accuracies indicating 95% confidence intervals over 600 tasks for *mini*Imagenet, and 2000 tasks for both *tiered*Imagenet and CUB, as is customary across the literature (Chen et al., 2019; Dhillon et al., 2020; Bateni et al., 2022). We compare our performance against a wide variety of state-of-the-art few-shot classification methods such as: (i) metric-learning (Wang et al., 2019; Bateni et al., 2020; Afrasiyabi et al., 2020; Yang et al., 2020), (ii) transductive feature-extraction based (Oreshkin et al., 2018; Ye et al., 2020; Li et al., 2019; Xu et al., 2021), (iii) optimization-based (Finn et al., 2017; Mishra et al., 2018; Oh et al., 2021; Lee et al., 2019; Rusu et al., 2019), (iv) transductive inference-based (Bateni et al., 2022; Boudiaf et al., 2020; Ziko et al., 2020; Liu et al., 2019), and (v) Bayesian (Iakovleva et al., 2020; Zhang et al., 2019; Hu et al., 2020; Patacchiola et al., 2020; Ravi & Beatson, 2019) approaches. Previous works such as Liu et al. (2019), and Hou et al. (2019) have demonstrated the superiority of transductive inference methods over their inductive counterparts. In this light, we compare against a larger number of transductive (18 baselines) rather than inductive (7 baselines) methods for a fair comparison. It is important to note that TRIDENT is only a *transductive feature-extraction* based method as we utilize the query set images to extract task-aware feature embeddings; it is not a transductive inference based method since we perform inference of class-labels over the entire domain of definition and not just for the selected query samples (Vapnik, 2006; Gammerman et al., 1998). The results on *mini*Imagenet and *tiered*Imagenet for both (5-way, 1 and 5-shot) settings are summarized in Table 2. We accentuate on the fact that we also compare against Transd-CNAPS+FETI (Bateni et al., 2022), where the authors pre-train the ResNet-18 backbone on the entire train split of Imagenet. We, however, avoid training on additional datasets, in favor of fair comparison with the rest of literature. Regardless of the choice of backbone (simplest in our case), TRIDENT sets a new state-of-the-art on *mini*Imagenet and *tiered*Imagenet for both (5-way, 1 and 5-shot) settings, offering up to 5% gain over the prior art. Recently, a more challenging *cross-domain* setting has been proposed for few-shot classification to assess its generalization capabilities to unseen datasets. The commonly adopted setting is

Table 2: Accuracies in (% ± std). The predominant methodology of the baselines: `Ind.`: inductive inference, `TF`: transductive feature extraction methods, `TI`: transductive inference methods. `Conv`: convolutional blocks, `RN`: `ResNet` backbone, †: extra data. Style: **best** and second best. `TRIDENT` employs a transductive feature extraction module (`TF`), and the simplest of backbones (`Conv4`).

| Methods | Backbone | Approach | *mini*Imagenet | | *tiered*Imagenet | | *mini*→CUB | |
| | | | 5-way 1-shot | 5-way 5-shot | 5-way 1-shot | 5-way 5-shot | 5-way 1-shot | 5-way 5-shot |
|---|---|---|---|---|---|---|---|---|
| MAML (Finn et al., 2017) | Conv4 | Ind. | 48.70 ± 1.84 | 63.11 ± 0.92 | 51.67 ± 1.81 | 70.30 ± 0.08 | 34.01 ± 1.25 | 48.83 ± 0.62 |
| ABML (Ravi & Beatson, 2019) | Conv4 | Ind. | 40.88 ± 0.25 | 58.19 ± 0.17 | - | - | 31.51 ± 0.32 | 47.80 ± 0.51 |
| OVE(PL) (Patacchiola et al., 2020) | Conv4 | Ind. | 48.00 ± 0.24 | 67.14 ± 0.23 | - | - | 37.49 ± 0.11 | 57.23 ± 0.31 |
| DKT+Cos (Patacchiola et al., 2020) | Conv4 | Ind. | 48.64 ± 0.45 | 62.85 ± 0.37 | - | - | 40.22 ± 0.54 | 55.65 ± 0.05 |
| BOIL (Oh et al., 2021) | Conv4 | Ind. | 49.61 ± 0.16 | 48.58 ± 0.27 | 66.45 ± 0.37 | 69.37 ± 0.12 | - | - |
| LFWT (Tseng et al., 2020) | RN10 | TF+TI | 66.32 ± 0.80 | 81.98 ± 0.55 | - | - | 47.47 ± 0.75 | 66.98 ± 0.68 |
| FRN (Wertheimer et al., 2021) | RN12 | Ind. | 66.45 ± 0.19 | 82.83 ± 0.13 | 71.16 ± 0.22 | 86.01 ± 0.15 | 54.11 ± 0.19 | 77.09 ± 0.15 |
| DPGN (Yang et al., 2020) | RN12 | TF+TI | 67.77 | 84.6 | 72.45 | 87.24 | - | - |
| PAL (Ma et al., 2021) | RN12 | TF+TI | 69.37 ± 0.64 | 84.40 ± 0.44 | 72.25 ± 0.72 | 86.95 ± 0.47 | - | - |
| Proto-Completion (Zhang et al., 2021a) | RN12 | TF+TI | 73.13 ± 0.85 | 82.06 ± 0.54 | 81.04 ± 0.89 | 87.42 ± 0.57 | - | - |
| TPMN (Wu et al., 2021) | RN12 | TF+TI | 67.64 ± 0.63 | 83.44 ± 0.43 | 72.24 ± 0.70 | 86.55 ± 0.63 | - | - |
| LIF-EMD (Li et al., 2021) | RN12 | TF+TI | 68.94 ± 0.28 | 85.07 ± 0.50 | 73.76 ± 0.32 | 87.83 ± 0.59 | - | - |
| Transd-CNAPS (Bateni et al., 2022) | RN18 | TF+TI | 55.6 ± 0.9 | 73.1 ± 0.7 | 65.9 ± 1.0 | 81.8 ± 0.7 | - | - |
| Baseline++ (Chen et al., 2019) | RN18 | TF | 51.87 ± 0.77 | 75.68 ± 0.63 | - | - | 42.85 ± 0.69 | 62.04 ± 0.76 |
| FEAT (Ye et al., 2020) | RN18 | TF | 66.78 | 82.05 | 70.80 | 84.79 | 50.67 ± 0.78 | 71.08 ± 0.73 |
| SimpleShot (Wang et al., 2019) | WRN | Ind. | 63.32 | 80.28 | 69.98 | 85.45 | 48.56 | 65.63 |
| Assoc-Align (Afrasiyabi et al., 2020) | WRN | TF | 65.92 ± 0.60 | 82.85 ± 0.55 | 74.40 ± 0.68 | 86.61 ± 0.59 | 47.25 ± 0.76 | 72.37 ± 0.89 |
| ReRank (SHEN et al., 2021) | WRN | TF+TI | 72.4±0.6 | 80.2±0.4 | 79.5±0.6 | 84.8±0.4 | - | - |
| TIM-GD (Boudiaf et al., 2020) | WRN | TI | 77.8 | 87.4 | 82.1 | 89.8 | - | 71 |
| LaplacianShot (Ziko et al., 2020) | WRN | TI | 74.9 | 84.07 | 80.22 | 87.49 | 55.46 | 66.33 |
| S2M2 (Mangla et al., 2020) | WRN | TF | 64.93 ± 0.18 | 83.18 ± 0.11 | 73.71 ± 0.22 | 88.59 ± 0.14 | 48.24 ± 0.84 | 70.44 ± 0.75 |
| MetaQDA (Zhang et al., 2021b) | WRN | TF | 67.83 ± 0.64 | 84.28 ± 0.69 | 74.33 ± 0.65 | 89.56 ± 0.79 | 53.75 ± 0.72 | 71.84 ± 0.66 |
| BAVARDAGE (Hu et al., 2022b) | WRN | TI | 82.7 | 89.5 | 83.5 | 89.0 | - | - |
| EASY (Bendou et al., 2022) | WRN | TF+TI | 84.04 ± 0.23 | 89.14 ± 0.11 | 84.29 ± 0.24 | 89.76 ± 0.14 | - | - |
| PT+MAP (Hu et al., 2021) | WRN | TF+TI | 82.92 ± 0.26 | 88.82 ± 0.13 | 85.67 ± 0.26 | 90.45 ± 0.14 | 62.49 ± 0.32 | 76.51 ± 0.18 |
| PE$M_n$E-BMS (Hu et al., 2022a) | WRN | TF+TI | 83.35 ± 0.25 | 89.53 ± 0.13 | 86.07 ± 0.25 | 91.09 ± 0.14 | 63.90 ± 0.31 | 79.15 ± 0.18 |
| Transd-CNAPS+FETI (Bateni et al., 2022) | RN18† | TF+TI | 79.9 ± 0.8 | 91.50 ± 0.4 | 73.8 ± 0.1 | 87.7 ± 0.6 | - | - |
| **TRIDENT(Ours)** | Conv4 | TF | **86.11 ± 0.59** | **95.95 ± 0.28** | **86.97 ± 0.50** | **96.57 ± 0.17** | **84.61 ± 0.33** | **80.74 ± 0.35** |

where one trains on *mini*Imagenet and tests on CUB (Chen et al., 2019). The results of this experiment are also presented in Table 2. We compare against *any existing baselines* for which this cross-domain experiment has been conducted. As can be seen, and to the best of our knowledge, `TRIDENT` again sets a new state-of-the-art by a significant margin of 20% for (5-way, 1-shot) setting, and 1.5% for (5-way, 5-shot) setting.

**Computational Complexity.** Most of the reported baselines in Table 2 use stronger backbones such as `ResNet12`, `ResNet18` and `WRN` which contain 11.5, 12.4 and 36.4 millions of parameters respectively. On the other hand, we use three `Conv4`s along with two fully connected layers and an `AttFEX` module which accounts for 410,958 and 412,238 parameters in the (5-way, 1-shot) and (5-way, 5-shot) scenarios, respectively. This is summarized in details in Table 3. Even though we are more parameter heavy than approaches that use a single `Conv4` as feature extractor, `TRIDENT`'s total parameters still lies in the same order of magnitude as these approaches. In summary, when it comes to complexity in parameter space, we are considerably more efficient than the vast majority of the cited competitors.

**Reliability Metrics.** A complementary set of metrics are typically used in probabilistic settings to measure the uncertainty and reliability of predictions. More specifically, expected calibration error (ECE) and maximum calibration error (MCE) respectively measure the expected and maximum binned difference between confidence and accuracy (Guo et al., 2017). This is illustrated in Table 4 where `TRIDENT` offers superior calibration on *mini*Imagenet (5-way, 1 and 5-shot) as compared to other

Table 3: Parameter count of `TRIDENT` against competitors.

| | Conv4 | $\mu_\phi$ | $\sigma_\phi$ | AttFEX | TRIDENT | Conv4 | RN18 | WRN |
|---|---|---|---|---|---|---|---|---|
| $q_{\phi_1}$ | 28896 | 51264 | 51264 | 6994 | **412,238** | 190,410 | 12.4M | 36.482M |
| $q_{\phi_2}$ | 28896 | 51264 | 51264 | - | | | | |
| $p_{\theta_1} + p_{\theta_2}$ | | 2245 + 132009 | | | | | | |

Table 4: Calibration errors of `TRIDENT`. Style: **best** and second best.

| | Metrics | MAML | PLATIPUS | ABPML | ABML | BMAML | VAMPIRE | TRIDENT |
|---|---|---|---|---|---|---|---|---|
| 5-way, 1-shot | ECE | 0.046 | 0.032 | 0.013 | 0.026 | 0.025 | 0.008 | **0.0036** |
| | MCE | 0.073 | 0.108 | 0.037 | 0.058 | 0.092 | 0.038 | **0.029** |
| 5-way, 5-shot | ECE | 0.032 | - | 0.006 | - | 0.027 | - | **0.0015** |
| | MCE | 0.044 | - | 0.030 | - | 0.049 | - | **0.018** |

Table 5: Style: **best** and second best.

| Methods | ECE | MCE | Brier |
|---|---|---|---|
| Feature Transfer(Chen et al., 2019) | 0.275 | 0.646 | 0.772 |
| Baseline(Chen et al., 2019) | 0.315 | 0.537 | 0.716 |
| Proto Nets(Snell et al., 2017) | **0.009** | 0.025 | 0.604 |
| DKT+Cos(Patacchiola et al., 2020) | 0.236 | 0.426 | 0.670 |
| BMAML+Chaser(Yoon et al., 2018) | 0.066 | 0.260 | 0.639 |
| LogSoftGP(ML)(Galy-Fajou et al., 2020) | 0.220 | 0.513 | 0.709 |
| LogSoftGP(PL)(Galy-Fajou et al., 2020) | 0.022 | 0.042 | 0.564 |
| OVE(ML)(Snell & Zemel, 2021) | 0.049 | 0.066 | 0.576 |
| OVE(PL)(Snell & Zemel, 2021) | 0.020 | 0.032 | 0.556 |
| **TRIDENT(Ours)** | **0.009** | **0.02** | **0.276** |

Table 6: Ablation study for *mini*Imagenet (5-way, 1-shot) tasks. Accuracies in ($\% \pm$ std.).

| (**B, n**) | (5, 3) | (5, 5) | | (10, 3) | (10, 5) | | (20, 3) | (20, 5) | |
|---|---|---|---|---|---|---|---|---|---|
| | - | $67.43 \pm 0.75$ | | $69.21 \pm 0.66$ | $74.6 \pm 0.84$ | | $80.82 \pm 0.68$ | $\mathbf{86.11 \pm 0.59}$ | |
| $(dim(\mathbf{z}_l),$ | (32, 32) | (32, 64) | (32, 128) | (64, 32) | (64, 64) | (64, 128) | (128, 32) | (128, 64) | (128, 128) |
| $dim(\mathbf{z}_c))$ | $76.29 \pm 0.72$ | $75.44 \pm 0.81$ | $79.1 \pm 0.57$ | $82.93 \pm 0.8$ | $\mathbf{86.11 \pm 0.59}$ | $85.62 \pm 0.52$ | $81.49 \pm 0.65$ | $82.89 \pm 0.48$ | $84.42 \pm 0.59$ |
| $(dim(\mathbf{W}_M),$ | (32, 32) | (32, 64) | (32, 128) | (64, 32) | (64, 64) | (64, 128) | (128, 32) | (128, 64) | (128, 128) |
| $dim(\mathbf{W}_N))$ | $78.4 \pm 0.23$ | $77.89 \pm 0.39$ | $79.55 \pm 0.87$ | $\mathbf{86.11 \pm 0.59}$ | $84.87 \pm 0.45$ | $82.11 \pm 0.35$ | $84.67 \pm 0.7$ | $85.8 \pm 0.58$ | $83.92 \pm 0.63$ |

probabilistic approaches, and MAML (Finn et al., 2017). To further examine the reliability and calibration of our method, we assess the ECE, MCE (Guo et al., 2017) and Brier scores (BRIER, 1950) of `TRIDENT` on the challenging *cross-domain* scenario of *mini*Imagenet $\rightarrow$ CUB for (5-way, 5-shot) tasks. When compared against other baselines that report these metrics on the aforementioned scenario, `TRIDENT` proves to be the most calibrated with the best reliability scores. This is shown in Table 5.

## 5.2 Decoupling Analysis

As a qualitative demonstration, we visualize the *label* and *context* latent means ($\mu_l$ and $\mu_c$) of query images for a randomly selected (5-way, 5-shot) task from the test split of *mini*Imagenet, before and after the MAML meta-update procedure. The `UMAP` (McInnes et al., 2018) plots in Fig. 5 illustrate significant improvement in class-conditional separation of query samples for *label* latent space upon meta-update, whereas negligible improvement is visible on the context latent space. This is qualitative evidence that $\mathbf{Z}_l$ captures more class-discriminating information as compared to $\mathbf{Z}_c$. To substantiate this quantitatively, the clustering capacity of these latent spaces is also measured by the Davies-Bouldin score (DBI) (Davies & Bouldin, 1979), where, the lower the DBI score, the better both the inter-cluster separation and intra-cluster "tightness". Fig. 5 shows that the DBI score drops significantly more after meta-update in the case of $\mathbf{Z}_l$ as compared to $\mathbf{Z}_c$, indicating better clustering of features in the former than the latter. This aligns with the proposed decoupling strategy of `TRIDENT` and corroborates the validity of our proposition to put an emphasis on label latent information for the downstream few-shot tasks.

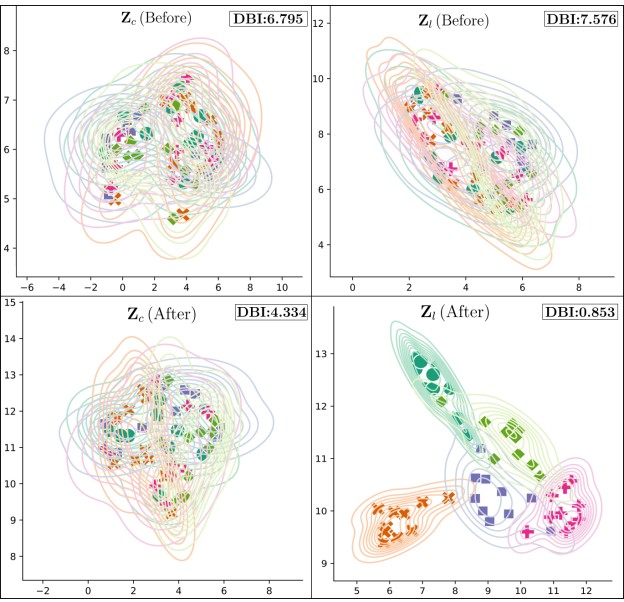

Figure 5: Better class separation upon meta-update is confirmed by lower DBI scores. Different colors/markers indicate classes.

## 5.3 Ablation Study

We analyze the classification performance of `TRIDENT` across various paramaters and hyper-parameters, as is summarized in Table 6. We use *mini*Imagenet (5-way, 1-shot) setting to carry out ablation study experiments. To cover different design perspectives, we carry out ablation on: (i) MAML-style training parameters: meta-batch size $B$ and number of inner adaption steps $n$, (ii) latent space dimensionality: $\mathbf{z}_l$ and $\mathbf{z}_c$ to assess the impact of their size, (iii) `AttFEX` features: number of features extracted by $\mathbf{W}_M$, $\mathbf{W}_N$. Looking at the results, `TRIDENT`'s performance is directly proportional to the number of tasks and inner-adaptation steps, as is previously demonstrated in Antreas Antoniou et al. (2019); Finn et al. (2017) for MAML based training. Regarding latent space dimensions, a correlation between a higher dimension of $\mathbf{z}_l$ and $\mathbf{z}_c$ and a better performance can be observed. Even though, the results show that increasing both dimensions beyond 64

leads to performance degradation. As such, $(64, 64)$ seems to be the sweet spot. Finally, on feature space dimensions of `AttFEX`, the performance improves when $\mathbf{W}_M > \mathbf{W}_N$, and the best performance is achieved when the parameters are set to $(64, 32)$. Notably, the exact set of parameters return the best performance for (5-way, 5-shot) setting. To sum up, $(B, n, dim(\mathbf{z}_l), dim(\mathbf{z}_c), dim(\mathbf{W}_M), dim(\mathbf{W}_N)) = (20, 5, 64, 64, 64, 32)$ turns out to be the best setting for (5-way, 1-shot), consistently the same for (5-way, 5-shot).

## 5.4 Impact of `AttFEX` and the Decoupled Inference Strategy

In order to study the impact of the transductive feature extractor `AttFEX`, we exclude it during training and train the remaining architecture. Training proceeds exactly as mentioned in Algorithm 2. As can be seen in Table 7, the exclusion of `AttFEX` from `TRIDENT` (`AttFEX` **OFF**) results in a substantial drop in classification performance across both datasets and task settings. Empirically, this further substantiates the importance of `AttFEX`'s ability to render the feature maps transductive/task-aware. As explained earlier in section 4.3, the derivation of `TRIDENT`'s ELBO implies that $y$ should be included as an input to $q_{\phi_1}$ due to its dependence on $\mathbf{z}_l$. However, in order to utilize `TRIDENT` as a classification and not a label reconstruction network, we choose not to input $y$ to $q_{\phi_1}(.)$, but rather do so indirectly by inducing a semblance of label characteristics in the features extracted from the images in a task. Thus, it is important to realize that this ability of `AttFEX` to render feature maps transductive is not just an adhoc performance enhancer, but rather an essential part of `TRIDENT`. To further understand the impact of `AttFEX` on `TRIDENT`, we train

Table 7: Impact of `AttFEX` on classification accuracies.

|  | *mini*Imagenet | | *tiered*Imagenet | |
|---|---|---|---|---|
|  | (5-way, 1-shot) | (5-way, 5-shot) | (5-way, 1-shot) | (5-way, 5-shot) |
| `AttFEX OFF` | $67.68 \pm 0.55$ | $78.53 \pm 0.21$ | $69.32 \pm 0.76$ | $79.32 \pm 0.76$ |
| `TRIDENT (EP)` | $69.84 \pm 0.5$ | $80.15 \pm 0.67$ | $73.29 \pm 0.60$ | $82.17 \pm 0.65$ |
| `TRIDENT (FEAT)` | $80.11 \pm 0.43$ | $87.61 \pm 0.12$ | $82.39 \pm 0.45$ | $88.78 \pm 0.39$ |
| `TRIDENT (LSTM)` | $75.41 \pm 0.49$ | $83.89 \pm 0.45$ | $79.72 \pm 0.52$ | $86.20 \pm 0.92$ |
| `ConvFEX` | $51.46 \pm 0.91$ | $62.35 \pm 0.72$ | $55.89 \pm 0.31$ | $64.56 \pm 0.29$ |
| `TRIDENT(Ours)` | $\mathbf{86.11 \pm 0.59}$ | $\mathbf{95.95 \pm 0.28}$ | $\mathbf{86.97 \pm 0.50}$ | $\mathbf{96.57 \pm 0.17}$ |

`TRIDENT` with a transductive feature extraction module different from `AttFEX`. The three modules that we replace `AttFEX` with are:

(i) Embedding propagation module (EP): This has been adapted from Embedding Propagation Networks (Rodríguez et al., 2020). Here, a non-parametric graph-based propagation matrix helps smoothen the embedding manifold to remove undesirable noise from the support and query feature vectors;

(ii) Attention-based feature adaption module (FEAT): This has been adapted from FEAT (Ye et al., 2020). A self-attention module is used to transform the support and query set by computing a weighted average of all the feature vectors in a task. The weights are calculated using a dot-product between each pair of feature vectors;

(iii) LSTM-based feature adaption module (LSTM): We introduce the LSTM-based transductive task-encoding procedure from Transductive CNAPS (Bateni et al., 2022) in place of `AttFEX` and carry out the same training procedure. The results for each of these experiments, when trained with `TRIDENT` on *mini*Imagenet and *tiered*Imagenet, are shown in Table 7.

`TRIDENT`'s superior results corroborate the importance of our design choices in `AttFEX`. Furthermore, to empirically verify the contribution of the decoupled variational inference vs `AttFEX`, we trained a simplified network `ConvFEX = Conv4 + AttFEX` as the inference network $q(\mathbf{z} | \mathbf{x})$ to generate class labels $y$ using an MLP $p(y | \mathbf{z})$. `ConvFEX` embodies the inference and generative mechanics of $\mathbf{z}_l$ while omitting the second latent variable $\mathbf{z}_c$, thus dropping the decoupled inference strategy. As shown in Table 7, the classification accuracies across both datasets and task settings for `ConvFEX` corroborate that when label-specific and context information are coupled, we observe a significant performance degradation as compared to `TRIDENT`, thus reaffirming the importance of our *decoupled* variational inference strategy.

## 5.5 Impact of End-to-End Meta-Learning

To understand the importance of end-to-end meta-training of the entire network architecture, we train parts of `TRIDENT` in different steps. More specifically, we pre-train a `ConvEnc` on the training split of *mini*Imagenet to perform 64-way classification. Note that during this pre-training phase, training proceeds by sampling random batches from the entire training split without defin-

ing support or query sets. We use the pre-trained feature extractors in `TRIDENT`'s inference networks $q_{\phi_1}$ and $q_{\phi_2}$ for fine-tuning. We then conduct three different experiments for fine-tuning the network: (i) freeze both the `ConvEnc`'s and fine-tune episodically without any MAML-style meta-learning; (ii) fine-tune the entire architecture episodically without any MAML-style meta-learning;

(iii) freeze both the `ConvEnc`'s and fine-tune using MAML-style meta-learning. Fine-tuning proceeds by sampling ($N$-way, $K$-shot) tasks from the training split of *mini*Imagenet. Notably, in (i) and (ii), we do not have separate updates for the support and query sets following simple episodic training. Therefore, employing an MLP for classification is a suboptimal utilization of the labelled samples. To address this, we use a prototypical classification framework as proposed in Prototypical Networks (Snell et al., 2017). The results of all the experimentation

Table 8: Impact of meta-learning on accuracies.

| | *mini*Imagenet | |
|---|---|---|
| | (5-way, 1-shot) | (5-way, 5-shot) |
| Frozen `ConvEnc` (Episodic) | $67.68 \pm 0.55$ | $78.53 \pm 0.21$ |
| Fine-tune `ConvEnc` (Episodic) | $69.84 \pm 0.5$ | $80.15 \pm 0.67$ |
| Frozen `ConvEnc` (Meta-Learn) | $80.11 \pm 0.43$ | $87.61 \pm 0.12$ |
| `TRIDENT`(Ours) | $\mathbf{86.11 \pm 0.59}$ | $\mathbf{95.95 \pm 0.28}$ |

is illustrated in Table 8. It can be observed that episodic fine-tuning is not as effective as meta-learning the entire network architecture. This can be attributed to the ability of MAML-style meta-learning to render the network's weights sensitive to the loss function, thus enabling quicker generalization to unseen tasks (Finn et al., 2017).

## 6 Concluding Remarks

We introduce a novel variational inference network (coined as `TRIDENT`) that simultaneously infers decoupled latent variables representing context and label information of an image. The proposed network is comprised of two intertwined variational sub-networks responsible for inferring the context and label information separately, the latter being enhanced using an attention-based transductive feature extraction module (`AttFEX`). Our extensive experimental results corroborate the efficacy of this transductive decoupling strategy on a variety of few-shot classification settings demonstrating superior performance and setting a new state-of-the-art for the most commonly adopted datasets *mini* and *tiered*Imagenet as well as for the recent challenging cross-domain scenario of *mini*Imagenet $\rightarrow$ CUB. As future work, we plan to demonstrate the applicability of `TRIDENT` in semi-supervised and unsupervised settings by including the likelihood of unlabelled samples derived from the graphical model. This would render `TRIDENT` as an all-inclusive holistic approach towards solving few-shot classification.

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
