# OpenReview forum: "Transductive Decoupled Variational Inference for Few-Shot Classification"
_TMLR — Accepted by TMLR_

### Review · Reviewer_Y8UK · 2022-12-23

**Summary Of Contributions:**

This work proposes a novel approach to few-shot learning for classification. In layman's terms, the proposed method is a hierarchical VAE in which the first latent variable $z_s$ provides semantic information (that information not important for classification), and the second latent variable $z_l$ embodies the information useful for classification. The authors make a number of modifications to make the model work in their settings, being the most important that:
1. The inference model does not depend on the label (so that it can be used in the during testing).
2. To cope with the loss of information, the authors shift to the transductive setting, using attention on query and support data *at once* to extract the relevant features (making it "task-aware").
3. Finally, the model is trained using meta-learning in the same fashion as MAML models.

These key changes enable a model which convincingly outperform existing methods using far less parameters than the average model, as empirically demonstrasted in their experiments.


**Audience:**

Yes

**Broader Impact Concerns:**

I don't have any concerns regarding the broader impact.

**Claims And Evidence:**

Yes

**Requested Changes:**

Critical changes to secure my recommendation:

1. The probabilistic model introduced in section 4.2 is misleading, in the sense that it is defined per sample $(x, y)$, but the proposed AttFEX introduces dataset-dependent dependences that are not contemplated anywhere. To my eyes, the probabilistic model should be defined in terms of the entire dataset $\mathcal{D}$, rather than individual points. While the generative network is inductive $p(\mathcal{D}|\mathcal{Z}) = \prod_i p(x_i | z_i)$, the inference network is transductive like $q(\mathcal{Z}|\mathcal{D}) = \prod_i q(z_i | \mathcal{D})$. This is a *huge* difference which is not reflected in the model. To help rewriting the probabilistic model, I would suggest looking at [Neural Processes](https://arxiv.org/abs/1807.01622), which also perform variational inference over the entire dataset.
2. Equation 2 is wrong, as the first KL should be inside an expectation w.r.t. $q_{\phi_2}(z_s|x)$. Also, that equation is the *negative* ELBO.
3. While I can understand $\alpha_1$ as the normalizing constant of a fixed variance Gaussian likelihood, $\alpha_2$ does not make sense in probabilistic terms and there is no such a thing as "being considered variational inference by consensus." Scaling factors such as those proposed can be understood instead as gradient-scaling parameters *during training*, see [Mitigating Modality Collapse in Multimodal VAEs via Impartial Optimization](https://arxiv.org/abs/2206.04496). This same paper also showed that properly tuning these parameters can improve the training in heterogeneous likelihoods (such as Gaussian + Categorical).
4. In section 5.4, the phrase "it is imperative to include y in the input to $q_{\phi_1}(.)$ for mathematical correctness of the variational inference formulation" is wrong. The choice of variational family is up to the user and will always be correct (another matter is wether it is feasible or sensible).

Changes that would strengthen the paper:

5. Formatting could be improved: properly using `\citep` and `\citet`, moving line numbers of algorithms outside the margins, remove the short-line at the beginning of page 11, correcting typos (e.g. remove doubled `equation equation` within the text), etc.
6. I'd make sure that the $86.11 \pm 0.59$ in Table 6 is correct, as it appears twice in the table.
7. I would mention meta-learning somewhere in the abstract, and definitely earlier than where it appears right now.
8. In problem definition, letter $K$ is used twice for difference things. Something similar goes for $\alpha$, $\alpha_1$ and $\alpha_2$. Indexes of $G_i$ after Equation 6 are wrong $1 \times C'$ instead of $C' \times 1$.
9. It is a bit odd to pass $X_{CE}^g$ as input in Algorithm 1, rather than computing it within the algorithm.

**Strengths And Weaknesses:**

**Strenghts:**
- The paper is clearly written and easy to follow for the most part. Figures help a lot to understand the proposed approach.
- The empirical results are strong and the qualitative results are convincing.
- Literature review is properly conducted, and the paper is mostly self-contained.
- The idea is intuitive, and all proposed changes are reasonable and well-argued.

**Weaknesses:**
- The definition of the probabilistic model has some issues that can be misleading (see in next section), as well as some claims in regards to the probabilstic methods are arguable.
- Limitations of the method are not properly discussed. For example, the model should not scale really well, as it needs to process the entire dataset at once.
- To my understanding, all results from other models are taken from the original papers. I would've appreciate to reproduce at least one of them with the actual codebase of this paper to make sure that the setups are equivalent.
- The last experiment (ConvFEX) in section 5.4 is unclear and (personally) unconvincing. Unclear in that I do not understand whether everything was exactly as in the proposed method (e.g., trained using meta-learning), as well as why to use a simplified model. Unconvincing as I would require a deeper discussion or stronger results to show a counter-intuitive result: if the only goal is classification, why wouldn't the model drop all irrelevant information? is it because reconstructing helps aligning the features of query and support variables?
- As far as I could see, there are no details regarding hyperparameter optimization.

---

> ### Author Response · Authors · 2023-01-23
> **Response to Reviewer Y8UK - Part 1**
>
> Thanks for your thorough review and insightful feedback on our paper. We are glad that you recognize the rationale behind our idea and that you find our qualitative results convincing. In the following, we will address your concerns, questions and recommendations in a point-to-point fashion.
> > Q1: The probabilistic model introduced in section 4.2 is misleading....
>
> A1: Thanks for the insightful and constructive remark. Based on your suggestion, we added a new Subsection (Subsection 4.4) that derives the negative ELBO of $\texttt{TRIDENT}$ incorporating our transductive inference network $q_\{\phi_1\}(\textbf{z}_l \mid \textbf{X}, \textbf{z}_s)$.
>
> Consider $\textbf{X} = \textbf{X}^{\mathcal{S}} \cup \textbf{X}^{\mathcal{Q}}$ to denote the tensor containing all images sampled in a task, $Y = Y^{\mathcal{S}} \cup Y^{\mathcal{Q}}$ to denote all the labels corresponding to the images in the task and $N'=NK+NQ$ to be the total number of samples in a task. Working with the IID assumption for all samples in a task, the likelihood of the entire task can be written as:
> $$
> \begin{aligned}
> p(\textbf{X}, Y) = \prod_{i=1}^{N'}\iint p(\textbf{x}_i, y_i \\,\vert\\, \textbf{z}_\{si\}, \textbf{z}_\{li\}) \\, p(\textbf{z}_\{si,\} \textbf{z}_\{li\}) \\, d \textbf{z}_\{si\} \\, d \textbf{z}_\{li\}.
> \end{aligned}
> $$
> Since the generative networks $p_\{\theta_\{2\}\}(\textbf{x} \\,\vert\\, \textbf{z}_\{s\}, \textbf{z}_\{l\})$ and $p_\{\theta_\{1\}\}(y \\,\vert\\, \textbf{z}_\{l\})$ remain inductive, while the approximate inference network $q_\{\phi_\{1\}\}\left(\textbf{z}_\{l\} \\,\vert\\, \textbf{X}, \textbf{z}_\{s\}\right)$ becomes transductive (via $\texttt{AttFEX}$), the log-likelihood now becomes:
> $$
> \begin{aligned}
> \ln p(\textbf{X}, Y) \geq \sum_\{i=1\}^{N'} \mathbb{E}_\{q_\{\phi_\{2\}\}\}\left[\mathbb{E}_\{q_\{\phi_\{1\}\}\}\left[\ln \left(\frac{p(\textbf{x}_i \\,\vert\\, \textbf{z}_\{si\}, \textbf{z}_\{li\}) p(y \\,\vert\\, \textbf{z}_\{li\}) p(\textbf{z}_\{s\}) p(\textbf{z}_\{l\})}{q(\textbf{z}_\{si\} \\,\vert\\, \textbf{x}_i) q(\textbf{z}_\{li\} \\,\vert\\, \textbf{X}, \\, \textbf{z}_\{si\})}\right)\right]\right].
> \end{aligned}
> $$
> Finally, the overall negative ELBO for the entire task can be given by:
> $$
> \begin{aligned}
> \mathcal{L}(\Psi) = -\sum_\{i=1\}^{N'} \\, \left[\mathbb{E}_\{q_\{\phi_\{2\}\}\}\mathbb{E}_\{q_\{\phi_\{1\}\}\}\left[\ln p_\{\theta_\{2\}\}(\textbf{x}_i \\,\vert\\, \textbf{z}_\{si\}, \textbf{z}_\{li\}) + \ln p_\{\theta_\{1\}\}(y_i \\,\vert\\, \textbf{z}_\{li\})\right] +
> \mathbb{E}_\{q_\{\phi_\{2\}\}\}\left[D_\{KL\}\big(q_\{\phi_\{1\}\}(\textbf{z}_\{li\} \\,\vert\\, \textbf{X}, \textbf{z}_\{si\}) \Vert\\, p(\textbf{z}_\{l\})\big)\right] +
> D_\{KL\}\big(q_\{\phi_\{2\}\}(\textbf{z}_\{si\} \\,\vert\\, \textbf{x}_i) \Vert\\, p(\textbf{z}_\{s\})\big)\right].
> \end{aligned}
> $$
> We chose to leave the derivation in Subsection 4.2 unaltered to maintain the flow of the manuscript since we do not explain our design choice to omit $y$ from $q_\{\phi_1}$ until Subsection 4.3.
> > Q2: Equation 2 is wrong, as...
>
> A2: Thanks for noticing this. Since we sample a random vector from the inferred Gaussian distribution using the reparameterization trick, in practice we do compute the expectation of the KL divergence $D_\{KL\}\big(q_\{\phi_\{1\}\}(\textbf{z}_\{l\} \\,\vert\, \textbf{x}, \textbf{z}_\{s\}) \Vert\\, p(\textbf{z}_\{l\})\big)$ with respect to $q_\{\phi_\{2\}\}(\textbf{z}_\{s\} \\,\vert\\, \textbf{x})$. However, as you suggested this is not reflected in the derivations, and we have now modified it according to your suggestion by $\mathbb{E}_\{q_\{\phi_\{2\}\}\}\left[D_\{KL\}\big(q_\{\phi_\{1\}\}(\textbf{z}_\{l\} \\,\vert\\, \textbf{x}, \textbf{z}_\{s\}) \Vert\\, p(\textbf{z}_\{l\})\big)\right]$. Also, ELBO has been changed to "negative ELBO'' in the sentence preceding Equation 2.
> > Q3: While I can understand $\alpha_1$ as the normalizing...
>
> A3: Thanks for this constructive remark. It helped us further improve the discussion around ELBO derivations. Notably, we mentioned this to clarify that even though the tightness of our derived ELBO is now altered, the framework would still be considered as variational inference by consensus among the literature. Following your suggestion, we now removed "...being considered variational inference by consensus" from the manuscript; and further promoted this as based gradient-scaling parameters, also citing the suggestion reference in our manuscript.
> > Q4: In section 5.4, the phrase "it is.....
>
> A4: Thanks for pointing this out, yet another constructive recommendation. To clarfiy, we mentioned this statement since the variable $y$ is dependent on $\textbf{z}_l$. However, we agree that the variational family can be modelled per the user's need, and thus, we have now modified the sentence following your suggestion.
> > Q5: Formatting could be improved...
>
> A5: We appreciate you noticing this. We have now ironed out these inconsistencies in formatting.

---

> ### Author Response · Authors · 2023-01-23
> **Response to Reviewer Y8UK - Part 2**
>
> > Q6: I'd make sure that the 86.11 +- 0.59 in ....
>
> A6: The number $86.11 \pm 0.59$ has only been mentioned once per ablation analysis in Table 6. We have double checked the table per your suggestion, and there are no mistakes!
> >Q7: I would mention meta-learning somewhere in the abstract...
>
> A7: Makes total sense. We have now included it in the abstract.
> > Q8: In problem definition, letter K is used...
>
> A8: In Section 3, we only mention $K$ to refer to the $K$-shots in a given task. We also explain that these $K$-shots signify the $K$-labelled samples per class in the support set, which refers again to the same thing. In short, we have not used $K$ to refer to two different things. $\alpha_1$ and $\alpha_2$ refer to the loss-scaling constants (already mentioned in Subsection 4.4) whereas $\alpha$ refers to the inner update’s learning rate (already mentioned in Algorithm 2). After Equation 6, it has already been mentioned that $G_i$ has a dimensionality of $1 \times C' \times W' \times H'$ and not $C' \times 1 \times W' \times H'$.
> >Q9: It is a bit odd to pass...
>
> A9: Thanks for the comment. This is a conscious choice to avoid confusing the reader between the $\texttt{ConvEnc}$ of $q_\{\phi_\{1\}\}$ and $q_\{\phi_\{2\}\}$. Since we already include the $\texttt{ConvEnc}$ of $q_\{\phi_\{2\}\}$ in line 2 of Algorithm 1, we choose to specify the $\texttt{ConvEnc}$ of $q_\{\phi_\{1\}\}$ in the text of Subsection 4.5 - Overview of $\texttt{TRIDENT}$ as "....the lower $\texttt{ConvEnc}$ block extracts feature maps  $\textbf{X}^{g}_\{\texttt{CE}\}=\texttt{ConvEnc}(\textbf{X}^{g})$".
>
> **Response to Comments from Weaknesses**
> > Q10: Limitations of the method are not...
>
> A10: Thanks for the constructive remark. In Section 5, Subsection - Implementational Details, we already mention a limitation of meta-learning $\texttt{TRIDENT}$ that is "Meta-learning objectives can lead to unstable optimization processes in practice, especially when coupled with stochastic sampling in latent spaces, as also previously observed in Antreas Antoniou et al., 2019". Following your suggestion, we now also specify the limitation regarding scaling $\texttt{AttFEX}$ to a larger number of samples in a task, in the same Subsection.
> > Q11: To my understanding, all results from ...
>
> A11: That is not entirely true. We have a mix of both. We reproduced a number of these references and where the difference in reproduction was negligible we have reported the same results as the authors. Of course, there are some references for which the full reproduction of the result would be a separate piece of work by itself. So, wherever we could reproduce even one main reported result, we have trusted the authors' claim and reported the rest. We use the exact datasets and splits provided by Ravi \& Larochelle, 2017a for $\textit{mini}$Imagenet, Ren et al., 2018 for $\textit{tiered}$Imagenet and Chen et al., 2019 for $\textit{mini}$Imagenet $\rightarrow$ CUB - which is most commonly adopted by a vast majority of the reported baselines. Here, in ResponseTable1, are a couple of examples of the reproduced results where the accuracies are in the same ballpark as the accuracies reported in the corresponding papers, thus, verifying the equivalence of training and testing setups.
>
> **ResponseTable1: Reproduced and Reported results.**
> |   Method                            | (5-way, 1-shot)  | (5-way, 1-shot)  | (5-way, 5-shot)  | (5-way, 5-shot)  |
> |------------------------------------|------------------|------------------|------------------|------------------|
> |                                    | Reproduced       | Reported         | Reproduced       | Reported         |
> | Laplacian Shot (Ziko et al., 2020) | 74.25 $\pm$ 0.15 | 74.9             | 84.87 $\pm$ 0.12 | 84.07            |
> | PT+MAP (Hu et al., 2021)           | 82.54 $\pm$ 0.73 | 82.92 $\pm$ 0.26 | 88.89 $\pm$ 0.43 | 88.82 $\pm$ 0.13 |

---

> ### Author Response · Authors · 2023-01-23
> **Response to Reviewer Y8UK - Part 3**
>
> > Q12: The last experiment (ConvFEX) in section 5.4 is unclear and (personally) unconvincing...
>
> A12: Thanks for the interesting question. The entire training strategy (MAML style meta-learning), the architecture of the $\texttt{ConvEnc}$ and the dimensionality of the inferred latent variable were kept the same when training $\texttt{ConvFEX}$, in order to have a fair comparison. The rationale behind removing the entire image-reconstruction VAE module from $\texttt{TRIDENT}$ was to test the impact of only inferring a single latent variable (practically merging the two decoupled variables into one) for image classification. The significant drop in performance corroborates the pivotal impact of the proposed decoupling strategy. Note that the starting sentence of Subsection 5.4 clarifies that the entire training strategy remains the same in the case of $\texttt{ConvFEX}$, $\textbf{\texttt{AttFEX} ON}$ and $\textbf{\texttt{AttFEX} OFF}$.
>
> **Response to** *`if the only goal is classification, why wouldn't the model drop all irrelevant information?'* - In an ideal scenario with unlimited data available per class, the model would indeed learn to drop all irrelevant information and only learn class-characterizing information for optimal performance. But in a $\textbf{few-shot}$ learning scenario, the model must learn to generalize quickly by identifying relevant information for classification only using a handful of samples since data available per class is scarce. In this case, injecting a handcrafted inductive bias of inferring both, class-relevant and irrelevant information, allows the network to understand and distinguish between the two types of information and only utilize the important class-characterizing information for classification.
> > Q13: As far as I could see, there are no details...
>
> A13: Thanks for the comment. The optimal hyperparameters were found using a grid search. We already mention the hyperparameters used for each setup and dataset in Section 5, Implementational Details.

---

> > ### Comment · Reviewer_Y8UK · 2023-01-27
> > **Response to the authors**
> >
> > Dear authors,
> > Thank you so much for the response.
> >
> > I appreciate the kind responses, specially A11: while my intuition was correct, my experience in few-shot learning is limited and I did not consider the setting the paper is framed in.
> >
> > After reading all the reviews and responses, I am happy to say that all my concerns were addressed, including my main concern (the probabilistic definition of the model, which has been properly clarified now).
> >
> > Congratulations.

---

> > > ### Author Response · Authors · 2023-01-30
> > > **Thanks**
> > >
> > > We thank you again for your constructive feedback.

---

### Review · Reviewer_MEaT · 2022-12-24

**Summary Of Contributions:**

The authors propose TRIDENT, a transductive few-shot learning method that has a generative model that disentangles label-relevant from label-irrelevant information in the latent space. To this end, they utilize a variational objective has a reconstruction term, which is based on MSE between the generated (via both latent semantic and label information) images and the original images, as well as a classification term, where the class label prediction is made from the label latent variables.

They also propose the AttFEX module, which is a transductive one that utilizes unlabeled information from both the support and query sets, in order to obtain a set of ‘task-cognizant’ features, which are used for the branch of the inference network that predicts the label-related latent variables. For this, they extract features from all images in the task using a convolutional feature extractor, and then utilize 1x1 convolutions for dimensionality reduction over (individual pixels of) all of the task’s images, thus blending information across classes. Next, they blend information across feature maps via a self-attention layer, the result of which is multiplied with the original non-transductive features (with a Hadamard product) to yield the transductive features. This is done independently for each of the support and query sets of the task.

Overall, they utilize a meta-learning framework to train the parameters in the convolutional feature extractors, AttFEX, and inference networks. Their meta-learning objective is a gradient-based one, where a few steps of gradient descent are performed on the support set to update all parameters (‘inner loop’), followed by an update on the query set (‘outer loop’). In both inner and outer loop, the transductive branch sees both the support and the query set (though without labels), whereas the non-transductive branch uses only the corresponding set (support set in inner loop, query set in outer loop).


**Audience:**

Yes

**Broader Impact Concerns:**

I have no broader impact concerns about this paper.


**Claims And Evidence:**

No

**Requested Changes:**

Clarity and correctness
==================
Metric learning is described as learning a “shared feature extractor to embed the samples into a metric space of class prototypes”. This is not correct, as building class prototypes is not a necessary part of all metric learning approaches. For example, out of the ones cited, there are several that don’t create prototypes, like Matching Networks which instead have a nonparameteric flavor. Further, the next sentence talks about how sharing a feature extractor across tasks isn’t necessarily ideal, but not all cited methods in this context share the feature extractor. E.g. Bateni et al (2020) conditions it using an amortized meta-learned model to output parameters for each new task, so this citation for instance should be part of the “task-aware” category instead.

Also, the definition of “task-aware” in the paper is an unconventional one that involves exploiting unlabeled data. This isn’t necessarily true - there is a vast literature of task-aware methods (also referred to as task-conditioning methods) where task-awareness is built based on the (labeled) support set of the task, like TADAM, CNAPs, Simple CNAPs etc. Perhaps a more appropriate term for what the authors are talking about here is “transductive task-aware”, since they are talking about a specific way of achieving task-awareness that makes additional assumptions (the availability of (additional) unlabeled data) over other task-aware methods.

In related work, SNAIL is incorrectly classified into the optimization-based meta-learning category. SNAIL is actually a black-box meta-learner that does not apply gradient descent at inference time to solve each new task, therefore it’s not optimization-based. Overall, from reading the intro and related work, it felt on several occasions like the authors haven’t read the papers they cited.

Please correct the citation style used throughout the paper which is incorrectly used and can be distracting. Use citep to refer to a paper and citet to refer to the authors of the paper. For instance, “Bateni et al. (2022) proposes a method for…” should use the latter, but “A method for … was proposed in (Bateni et al. 2022)” should use the former.

Style, design and context don’t sound like “semantic” information to me. The term semantics is typically used to refer to information relating to e.g. class information, so I find this terminology odd. A more appropriate term would be perhaps nuisance, style or context variables.

Problem definition: it should be noted that the splits into train/valid/test is done at a *class* level (assuming the authors are following the standard few-shot classification setup).

“The NQ query and NK support images are mutually exclusive” - disjoint is a more appropriate word than mutually exclusive, since we’re referring to sets.

Section 3 presents a framework for variational inference for a graphical model that assumes that images are generated from latent factors corresponding to both “semantics” and “label” information, whereas the latter is responsible for generating the label. However, it is unclear in this section how the loss function presented here fits into the episodic framework. The authors say that the loss is applied separately to both the support and query sets which makes me wonder then, why is it useful to have a support/query separation? The role of the support and query becomes clear later but at this stage causes confusion. I would recommend not mentioning support/query in this section and only describing the variational inference model there.

Do the two feature extractors shown in figure 4 have shared parameters with each other? The use of different colors in this figure made me guess that they don’t, but that would be unintuitive and perhaps wasteful. If they don’t have shared parameters, it would be good to include an empirical comparison to a version that does share them.

I did not understand the difference between transductive feature extraction and transductive inference. It would be good to explain this in more detail.

In Figure 5 for the visualization, it should be clarified whether the query images used to create this were from the meta-training or meta-validation/meta-testing set. Also, in the paragraph describing this figure (section 5.2), the authors introduce the term ‘meta-adaptation’, while the caption of Fig. 5 calls this ‘meta-update’, neither of which has been defined. I assume this means task-adaptation using the support set, but this needs to be clarified. Note that, if my assumption is correct, the terminology used here is confusing since in meta-learning, ‘meta-update’ refers to the update to the meta-parameters, which is based on the query set loss. Either way, meta-adaptation and meta-update are not synonyms.

Minor: in equation 4, maybe use a symbol other than N to denote the result of the 1x1 convolution, since N denotes the number of classes in each task.


Experimental setup
===============
It would be useful to also have ablations with no meta-learning, or less meta-learning, as these would better aid in understanding the success of the proposed approach. This is important as the proposed system is quite complex and involves several components, and it’s not clear whether learning all of them need to be end-to-end meta-learned, or what is the effect of that design choice. For instance, can one pre-train the convolutional feature extractor and inference network for the variational objective to reconstruct images, using batches of examples from the meta-training set, without separate support/query sets? Then this could be kept frozen during the meta-learning phase which can learn e.g. the AttFEX parameters. Pushing further along this direction, could one use a pre-trained convolutional feature extractor (e.g. trained on ImageNet) and integrate that (frozen) into this architecture? Some experiments in this direction would strengthen the paper.

It would also be great to have other ablations for the transductive component, aside from the current ablation that removes it entirely. What are the architectures used in the cited related works to handle incorporating unlabeled information from the entire task? Can one of these be subbed in in the place of AttFEX?

The experimental setting is focused on older benchmarks (and thus also precludes comparisons against newer methods). The community has largely moved away from simpler benchmarks like mini- and tiered- imagenet that are relatively homogeneous and evidently don’t require adaptation to the feature extractor for good performance on new few-shot tasks. While the cross-domain scenario explored here is useful, it is quite limited. It would be much more informative to instead use more recent benchmarks for few-shot learning like Meta-Dataset [1], Meta-Dataset + VTAB [2], Orbit [3], MetaAlbum [4], etc.

References
=========
[1] Meta-Dataset: A dataset of datasets for learning to learn from few examples. Triantafillou et al. ICLR 2020.

[2] Comparing Transfer and Meta Learning Approaches on a Unified Few-Shot Classification Benchmark. Dumoulin et al. NeurIPS 2021.

[3] ORBIT: A Real-World Few-Shot Dataset for Teachable Object Recognition. Massiceti et al. ICCV 2021.

[4] Meta-Album: Multi-domain Meta-Dataset for Few-Shot Image Classification. NeurIPS 2022.


**Strengths And Weaknesses:**

Strengths:
========
- the idea of disentangling latent factors in terms of those relating to class labels and those that don’t is a good one, as indeed utilizing only the former set for classification can intuitively be a better fit for few-shot classification problems due to focusing on a more ‘robust’ set of features, which is important in the face of limited task-specific data.

- the proposed method performs very strongly on the problems it is tested on, which is particularly impressive given that it uses a much smaller architecture compared to other works they compare against.

Weaknesses:
===========
- the paper has several clarity and writing issues

- the proposed architecture is more complicated than some of the previous work compared against, which makes it harder to understand and deconstruct the improved performance over previous work. While the authors perform some ablations which are helpful, they are not sufficient. For instance, they ablate entirely removing the AttFEX module, but don’t consider simpler architectural designs for that module.

- in terms of experimental setup, the paper focuses on simple benchmarks that the community is largely moving away from, in favor of more diverse few-shot learning benchmarks.

---

> ### Author Response · Authors · 2023-01-23
> **Response to Reviewer MEaT - Part 1**
>
> Thank you for your detailed review and insightful feedback on our paper. We are glad that you find our core idea of inferring decoupled latent variables intuitive and our empirical results impressive. In the following, we will address your concerns, questions and recommendations in a point-to-point fashion.
> > Q1: Metric learning is described as learning ...
>
> A1: Thanks for your intriguing comment. We agree that building class prototypes might not be a necessity for metric learning approaches. But in practice, almost all metric learning approaches explicitly or implicitly compute the mean feature vector using support-set samples of each class, which in turn is called "class prototype".
> Exceptionally, Matching Networks are limited to only one labelled sample per class; thus, the authors do not need to explicitly compute the mean feature vector per class.
> It might also seem that Relation Networks (also cited in our manuscript - Sung et al., 2018) do not build class prototypes, however, they implicitly do so by summing over the support feature vectors of each class, which is just a scaled version of computing the class mean, aka prototypes. All the other methods cited in that sentence in Section 1 (Prototypical Nets, SimpleShot and Prototype Rectification) also explicitly build class prototypes by computing the mean feature vector per class.
>
> We would also like to highlight that we do not specifically mention class prototypes in Section 2 where we discuss Metric Learning in detail. We only do so in Section 1 to point out a limitation of most Metric Learning approaches.
> Regarding the citation of Bateni et al., (2020) under Metric Learning approaches, we agree that they indeed use a $\texttt{ResNet}$ backbone which is enhanced with $\texttt{FiLM}$ layers that condition the model parameters to all support samples in a task. We included this in the metric learning approaches since the feature extractor utilizes only the support samples and not all samples in a task (support and query). However, following your comment on "task-aware" vs "transductive task-aware", we have removed it from the Metric Learning category.
> > Q2: Also, the definition of “task-aware” in the paper....
>
> A2: Thanks for the insightful comment. We totally agree and following your suggestion, we have changed "task-aware" to "transductive task-aware" when referring to our approach and in the title of Section 2. We would also like to highlight the fact that we already mention $\texttt{AttFEX}$ as a "transductive feature extraction" method in the title of Subsection 4.3.
> > Q3: In related work, SNAIL is incorrectly...
>
> A3: Thanks for noticing and raising this. SNAIL is indeed not a gradient-based optimization method. However, we included it in the optimization-based meta-learning category due to its use of temporal convolutions and causal attention layers to have access over multiple past timesteps. Following your suggestion, we have now replaced this with Meta-Learner LSTM [1].
> We would like to accentuate that we have put a lot of thought and effort into carefully hand-picking these references and creating this categorization. Apart from SNAIL (which now modified according to your suggestion in categorization), we believe we have done justice in properly categorizing the prior arts in suitable divides.
> > Q4: Please correct the citation style used ....
>
> A4: Thanks for noticing. We have now adjusted the few instances with an incorrect citation style, per your suggestions.
> >Q5: Style, design and context don’t sound like “semantic” ....
>
> A5: Thanks for the suggestion. The term "context'' is most often used when referring to the information learnt at the topmost hierarchy in a hierarchical latent variable model. Since this is quite different from the way we use our "semantic'' latent variable, we chose not to use the word "context" to avoid confusing the reader. The term "style" is rather specific and would be appropriate for smaller, more limited datasets where there are multiple images of the same class with style being the only factor of variation. $\textit{mini}$Imagenet and $\textit{tiered}$Imagenet are subsets of Imagenet that contain different images across different classes, not just differentiated by style but also by several other factors such as angle of capture, scenery, colour, positioning etc. Thus, to avoid confusing the reader we chose the term "semantic" to address the other latent variable. That said, among the suggested options we feel "context'' could still be a good replacement naming convention and we are open to modifying "semantic'' with "context'', if you strongly recommend doing so. Please let us know.
> > Q6: Problem definition: it should ...
>
> A6: Indeed, we abide by the standard few-shot classification setup where the splits are governed by the classes. Details regarding datasets and their splits have already been specified in Section 5, Subsection on "Benchmark Datasets''.

---

> ### Author Response · Authors · 2023-01-23
> **Response to Reviewer MEaT - Part 2**
>
> > Q7: “The NQ query and NK support ...
>
> A7: That is a good suggestion, and we agree. We have modified the text accordingly.
> > Q8: Section 3 presents a framework for variational inference for a ...
>
> A8: We mention that the loss is calculated for support and query sets separately because we follow the meta-learning strategy as proposed in MAML (Finn et al., 2017). Here, the inner updates are computed on the support sets and the meta updates are on the query sets. This detail has already been discussed in Subsection 4.2 and has been explained at length in Subsection 4.4, Algorithm 2. We, however, understand that it still might cause confusion as you suggested; thus, we have removed this line from Subsection 4.2 for improved clarity.
> > Q9: Do the two feature extractors shown in figure 4 ...
>
> A9: Thanks for your thorough observations and constructive remarks. The two feature extractors shown in Figure 4 do not have shared parameters, and thus, have different colours - as you rightfully observed. The roles of both $\texttt{ConvEnc}$s are different; as a result, the choice of not sharing the weight parameters of both feature extractors is conscious and the design. The primary role of the $\texttt{ConvEnc}$ of $q_\{\phi_\{1\}\}(.)$ is to extract class-relevant feature maps from images that help improve the classification performance of the network. The $\texttt{ConvEnc}$ of $q_\{\phi_\{2\}\}(.)$ must extract feature maps that are not only useful for image reconstruction but must also help with the inference of the label-latent vector $\textbf{z}_l$. Following your point, and to further verify this empirically, we trained a modified version of $\texttt{TRIDENT}$ with shared weights between the two $\texttt{ConvEnc}$’s on $\textit{mini}$Imagenet. Note that apart from learning-rate tuning, in this new experiment, all the other design choices and settings remain the same for a fair comparison. The results of this experiment are given in ResponseTable2 below:
>
> **ResponseTable2: Shared $\texttt{ConvEnc}$s**
> |      Methods                              | (5-way, 1-shot)         | (5-way, 5-shot)        |
> |------------------------------------|-----------------------------|----------------------------|
> | $\texttt{TRIDENT}$(shared ConvEncs) | 65.72 $\pm$ 0.23            | 72.90 $\pm$ 0.56           |
> | $\texttt{TRIDENT}$                 | $\textbf{86.11 $\pm$ 0.59}$ | $\textbf{95.95 $\pm$ 0.28}$ |
>
> The significant drop in performance of $\texttt{TRIDENT}$(shared  $\texttt{ConvEnc}$) as compared to $\texttt{TRIDENT}$ confirms the need for using separately trainable parameters for both the $\texttt{ConvEnc}$s. We will add this to the revised manuscript.
> > Q10: I did not understand the difference ....
>
> A10: Transductive feature extraction refers only to the extraction of features from images using both, the labelled (support) and unlabelled (query) samples (Bateni et al., 2022). Transductive inference on the other hand refers to performing classification/inference only on the given unlabelled samples (query set) all at once, rather than learning a decision boundary on the entire embedding space (Vapnik, 2006; Gammerman et al., 1998). For example, in TPN (Liu et al., 2019), the labels are propagated from the support set to the unlabelled query set only and all at once. In contrast to this, if an MLP were used to obtain logits for query samples, a function would be learnt that potentially classifies every point present in the domain of the embedding space, and not just the given unlabelled samples. That said, following your remark, we have further elaborated on this in Subsection 5.1.
> > Q11: In Figure 5 for the visualization, it ...
>
> A11: Thanks for the constructive remark. We randomly sample this task from the test set of $\textit{mini}$Imagenet. We now include this information as an additional sentence in Subsection 5.2. The figure illustrates the latent variables before and after the meta-update procedure, as already explained in Algorithm 2. To improve the clarity of this analysis, we have replaced the term ‘meta-adaptation’ with ‘meta-update’, which refers to the MAML-style outer update of the network’s parameters.

---

> ### Author Response · Authors · 2023-01-23
> **Response to Reviewer MEaT - Part 3**
>
> > Q12: It would be useful to also have ablations with no ...
>
> A12: Thanks for the remark. In order to understand the role of meta-learning in training $\texttt{TRIDENT}$, in Table 6, we already conduct experiments where, in a way, we vary the degree of meta-learning when training  $\texttt{TRIDENT}$. In the table, the first 2 rows show the ablation of $(B,n)$ which are the meta-batch size (number of tasks on which inner updates are computed) and the number of inner updates in the meta-learning procedure, respectively. We observed that the performance increases as both these parameters increase since there is a greater amount of data to learn from and an increased number of iterations to update weights.
>
> More importantly, following your remarks, we have conducted two more experiments on ($5$-way, $1$-shot) $\textit{mini}$Imagenet tasks with an increased number of inner update steps to strengthen our understanding of its role. The results are shown below in ResponseTable3:
>
> **ResponseTable3: Ablation of ($B$, $n$)**
> | ($B$,$n$)          | (20, 7)      | (20, 10)     |
> |--------------------|------------------|------------------|
> | $\texttt{TRIDENT}$ | 78.23 $\pm$ 0.98 | 54.91 $\pm$ 0.77 |
>
> The performance drops drastically when the number of inner updates increases beyond $5$. This suggests that after a certain threshold of $n$ ($5$ in this setting), the network parameters start to overfit on the support sets of the sampled tasks, thus leading to inferior performance.
>
> **Response** to *End-to-end training and separate training modules*: Thanks for the insightful remark. Pre-training a $\texttt{ConvEnc}$ and the inference networks $\mu_\{\phi\}(.)$, $\sigma^2_\{\phi\}(.)$ would only be possible for the network that infers $\textbf{z}_s$ and not $\textbf{z}_l$. Since the inference networks of $\textbf{z}_l$ ($\mu_\{\phi_\{1\}\}(.)$, $\sigma^2_\{\phi_\{1\}\}(.)$) also require a latent vector from $\textbf{z}_s$ as input, we would need to alter the architecture of the inference networks at fine-tuning, which is not ideal. However, the second suggestion is feasible and based on this, we have now pre-trained a $\texttt{ConvEnc}$ on the training split of $\textit{mini}$Imagenet to perform $64$-way classification. Note that during this pre-training phase, training proceeds by sampling random batches from the entire training split without defining support or query sets. We use the pre-trained feature extractors in $\texttt{TRIDENT}$'s inference networks $q_\{\phi_\{1\}\}$ and $q_\{\phi_\{2\}\}$ for fine-tuning. In order to understand the impact of meta-learning in training $\texttt{TRIDENT}$, we conduct three experiments for fine-tuning the network:
> 1. Freeze both the $\texttt{ConvEnc}$'s and fine-tune episodically (without any MAML-style meta-learning).
> 2. Fine-tune the entire architecture episodically (without any MAML-style meta-learning).
> 3. Freeze both the $\texttt{ConvEnc}$'s and fine-tune using MAML-style meta-learning.
>
> Fine-tuning proceeds by sampling ($N$-way, $K$-shot) tasks from the training split of $\textit{mini}$Imagenet. In the case of fine-tuning using simple episodic training and not MAML-style meta-learning, we do not have separate updates for the support and query sets. Due to this, employing an MLP for classification is a sub-optimal utilization of the labelled samples and thus we use a prototypical classification framework as proposed in Prototypical Networks (Snell et al., 2017). The results of all the proposed experiments have been shown below in ResponseTable4.
>
> **ResponseTable4: Separate Training Modules**
> | Methods                                 | (5-way, 1-shot)             | (5-way, 1-shot)             |
> |-----------------------------------------|-----------------------------|-----------------------------|
> | Frozen $\texttt{ConvEnc}$ (Episodic)    | 63.78 $\pm$ 0.79            | 69.34 $\pm$ 0.90            |
> | Fine-Tune $\texttt{ConvEnc}$ (Episodic) | 71.72 $\pm$ 0.28            | 76.40 $\pm$ 0.59            |
> | Frozen $\texttt{ConvEnc}$ (Meta-Learn)  | 78.23 $\pm$ 0.83            | 84.45 $\pm$ 0.32            |
> | $\texttt{TRIDENT}$                      | $\textbf{86.11 $\pm$ 0.59}$ | $\textbf{95.95 $\pm$ 0.28}$ |
>
> It can be observed that episodic fine-tuning is not as effective as meta-learning the architecture. This can be attributed to the ability of MAML-style meta-learning to render the network's weights sensitive to the loss function, thus enabling quicker generalization to unseen tasks (Finn et al., 2017). In summary, we observe that having separate pre-training and fine-tuning modules for a part of the entire network architecture leads to significantly inferior performance as compared to our end-to-end meta-learning strategy, thus corroborating its importance. We think this is an interesting experiment, and thus, we have added this to a new Subsection (Subsection 5.5) in the manuscript.

---

> ### Author Response · Authors · 2023-01-23
> **Response to Reviewer MEaT - Part 4**
>
> > Q13: It would also be great to have other ablations for ...
>
> A13: Thanks for the constructive remark. We conducted 3 additional experiments with a transductive feature extraction module different from $\texttt{AttFEX}$. The 3 modules have been listed below:
> 1. Embedding propagation module (EP): This has been adapted from Embedding Propagation Networks [7]. Here, a non-parametric graph-based propagation matrix helps smoothen the embedding manifold to remove undesirable noise from the support and query feature vectors.
> 2. Attention-based feature adaption module (FEAT): This has been adapted from FEAT (Ye et al., 2020). A cross-attention module is used to transform the query set by computing a weighted average of the support set feature vectors. The weights are calculated using dot-product similarity between a given query-set vector and all the support-set feature vectors.
> 3. LSTM-based feature adaption module (LSTM): We introduce the LSTM-based transductive task-encoding procedure from Transductive CNAPS (Bateni et al., 2022) in place of $\texttt{AttFEX}$ and carry out the same training procedure.
>
> The results for each module, when trained with $\texttt{TRIDENT}$ on $\textit{mini}$Imagenet and $\textit{tiered}$Imagenet, are shown below in ResponseTable5:
>
> **ResponseTable5: Replacing $\texttt{AttFEX}$**
> | Methods                    | $\textit{mini}$Imagenet     | $\textit{mini}$Imagenet     | $\textit{tiered}$Imagenet   | $\textit{tiered}$Imagenet   |
> |----------------------------|-----------------------------|-----------------------------|-----------------------------|-----------------------------|
> |                            | (5-way, 1-shot)             | (5-way, 5-shot)             | (5-way, 1-shot)             | (5-way, 5-shot)             |
> | $\texttt{TRIDENT}(EP)$     | 69.84 $\pm$ 0.5             | 80.15 $\pm$ 0.67            | 73.29 $\pm$ 0.60            | 82.17 $\pm$ 0.65            |
> | $ \texttt{TRIDENT}(FEAT) $ | 79.11 $\pm$ 0.43            | 87.61 $\pm$ 0.12            | 83.39 $\pm$ 0.45            | 88.78 $\pm$ 0.39            |
> | $ \texttt{TRIDENT}(LSTM) $ | 75.41 $\pm$ 0.49            | 81.89 $\pm$ 0.45            | 79.72 $\pm$ 0.52            | 86.20 $\pm$ 0.92            |
> | $ \texttt{TRIDENT} $       | $\textbf{86.11 $\pm$ 0.59}$ | $\textbf{95.95 $\pm$ 0.28}$ | $\textbf{86.97 $\pm$ 0.50}$ | $\textbf{96.57 $\pm$ 0.17}$ |
>
> Again, thanks for this constructive feedback. We now have more evidence to add to the manuscript (Subsection 5.4, Impact of $\texttt{AttFEX}$) corroborating the importance of our design choice and their impact.
> > Q14: The experimental setting is focused on older ...
>
> A14: While it is true that $\textit{mini}$Imagenet and $\textit{tiered}$Imagenet are older benchmarks, these are still the most widely adopted benchmarks. We intentionally chose to evaluate $\texttt{TRIDENT}$ on these benchmarks in order to be able to compare against the vast majority of prior art and state-of-the-art on few-shot learning. Even the most recent (almost all 2021-2022) state-of-the-art methods for few-shot learning such as Transductive CNAPS (Bateni et al., 2022), BAVARDAGE [3], PEMnE-BMS (Hu et al. 2022), PT+MAP (Hu et al., 2022), EASY [4], ASY [4], SOT [2], Sill-Net [5] and HCTransformers [6] evaluate on both $\textit{mini}$Imagenet and $\textit{tiered}$Imagenet.
> Notably, as a side information, none of the studies we compare against benchmark their performance on MetaAlbum and Orbit. Per your suggestion, we tried to setup experiments on Meta-Dataset and it turns out that since sampled episodes would randomly contain anywhere between $5$ to $50$ number of classes, and $1$ to $200$ number of shots, it would become infeasible for our compute resources (11GB NVIDIA 1080Ti GPU's) to operate with these larger number of samples per task. As such, we unfortunately have to set this interesting direction for our future studies.
>
> **References**:
>
> [1] Ravi, Sachin, and Hugo Larochelle. "Optimization as a model for few-shot learning." (2016).
>
> [2] Shalam, Daniel, and Simon Korman. "The Self-Optimal-Transport Feature Transform." arXiv preprint arXiv:2204.03065 (2022).
>
> [3] Hu, Yuqing, Stéphane Pateux, and Vincent Gripon. "Adaptive Dimension Reduction and Variational Inference for Transductive Few-Shot Classification." arXiv preprint arXiv:2209.08527 (2022).
>
> [4] Bendou, Yassir, et al. "EASY: Ensemble Augmented-Shot Y-shaped Learning: State-Of-The-Art Few-Shot Classification with Simple Ingredients." arXiv preprint arXiv:2201.09699 (2022).
>
> [5] Zhang, Haipeng, et al. "Sill-net: Feature augmentation with separated illumination representation." arXiv preprint arXiv:2102.03539 (2021).
>
> [6] He, Yangji, et al. "Attribute Surrogates Learning and Spectral Tokens Pooling in Transformers for Few-shot Learning." Proceedings of the IEEE/CVF Conference on Computer Vision and Pattern Recognition. 2022.
>
> [7] Rodríguez, Pau, et al. "Embedding propagation: Smoother manifold for few-shot classification." ECCV, 2020.

---

> > ### Comment · Reviewer_MEaT · 2023-02-01
> > **response to authors**
> >
> > Thank you for the response to my review.
> >
> > I disagree with the comments about metric learning here again. Metric learning is a very broad area of research that can’t be summarized as “metric learning proposes to learn a shared feature extractor to embed the samples into a metric space of class prototypes” – this is the 3rd sentence in the introduction in the current draft, which is incorrect.
> > It's a coincidence that the papers the authors chose to cite all compute prototypes but this certainly isn't the case in general. Note that Matching Networks don't only work for 1-shot. In fact, you will see performance for this method being reported e.g. for 5-shot settings too in most few-shot learning papers (see e.g. the Prototypical Network paper’s tables of results) and more generally for variable numbers of shots in benchmarks like Meta-Dataset. While Prototypical Networks and Matching Networks are indeed equivalent for 1-shot, they certainly aren't for a number of shots > 1 (again, the latter never computes prototypes). Other important metric learning approaches that don't compute prototypes in any way are e.g. siamese networks (Koch et al), triplet networks (Hoffer et al), and other works that make example-wise comparisons, like mAP-SVM/mAP-DLM (Triantafillou et al) which is specifically applied to few-shot learning.
> >
> > Thank you for the ablations of not sharing the weights of the two feature extractors, not performing maml-like updates, using pretrained (either frozen or finetuned) feature extractors to investigate the effect of episodic training and meta-learning, and using different modules in place of AttFEX. These strengthen the paper’s conclusions significantly.
> >
> > Thank you for all other clarifications. Regarding terminology, I see your points about not using context or style. What about ‘nuisance variables’ vs ‘label variables’? This isn’t hugely important either way but I think it would really help readability, as the word ‘semantics’ is usually linked to high-level concepts (like those actually related to class labels!) and not to label-agnostic information.
> >
> > References
> > - Siamese Networks for One-shot Image Recognition. Koch et al.
> > - Deep Metric Learning using Triplet Network. Hoffer et al.
> > - Few-shot Learning through an Information Retrieval Lens. Triantafillou et al.

---

> > > ### Author Response · Authors · 2023-02-06
> > > **Response to Reviewer MEaT - Part 4.1**
> > >
> > > Thanks for the constructive feedback.
> > >
> > > > Q15. I disagree with the comments about metric learning ...
> > >
> > > A15: We understand that the three mentioned metric learning references of Koch et al., Hoffer et al. and Triantafillou et al. do not specifically build class prototypes by computing the mean feature vector using support-set samples of each class. Following your suggestion, we have now modified this sentence in Section 1 to discuss _aggregated class embeddings_ instead of the class prototypes in a learnt metric space.
> > >
> > > > Q16. Thank you for the ablations....
> > >
> > > A16: We are glad that our additional experiments are convincing to substantiate our claims and elevate your concerns.
> > >
> > > > Q17: Thank you for the clarifications ...
> > >
> > > A17: Thanks for the comment. Following your suggestion, we propose to use the word _context_ instead of _semantic_ to address the latent variable $\textbf{z}_s$ throughout our work. We think that the word "nuissance" might be more distracting than meaningful for the readers. Following this change, we think it would be more appropriate to use the term $\textbf{z}_c$ instead of $\textbf{z}_s$ for the **c**ontext latent variable.
> > >
> > > All these proposed changes have been highlighted in $\textcolor{blue}{blue}$ in the updated manuscript.

---

### Review · Reviewer_sHKr · 2023-01-11

**Summary Of Contributions:**

The paper proposes a technique for few-shot learning by jointly learning semantic and label latent representations.

**Audience:**

Yes

**Claims And Evidence:**

Yes

**Requested Changes:**

- I do not understand what decoupled stands for in the title and throughout the paper. How comes $z_s$ and $z_l$ are decoupled if $z_l$ clearly depends on $z_s$ in Figure 2 and equation (1)? Please fix this.
- Equation deriving ELBO uses factorization of $p(z_s, z_l)$ assuming $z_s$ and $z_l$ are independent. First, what is the justification for this assumption? Is it really possible that label is independent of semantic information? It is hardly believable that this independence is actually viable in practice. Since this is foundational to the theory of the paper, I believe it is very important to motivate theoretically and empirically that this assumption holds in practice or otherwise modify the theory accordingly so that the algorithm does not need to rely on this assumption. Second, Figure 1 does show the arrow conditioning $z_l$ on $z_s$, which again contradicts the theoretical foundation of the paper.
- Related work fails to demonstrate where the current work falls in the literature and which research gaps it helps to bridge. Please rewrite accordingly.
- Relationship and novelty w.r.t. matching networks and other few-shot approaches using attention must be more clearly articulated
- "This helps the network utilize information across corresponding pixels of all images in a task $T_i$ which acts as a parametric comparison of classes". Can you provide empirical support of this claim in the manuscript?
- "This increases the sensitivity of the parameters Ψ towards the loss function for fast adaptation to unseen tasks and reduces generalization errors on the query set $Q$." Can you provide empirical support of this claim in the manuscript?
- The main idea and claim of the paper is "Thus, we argue that attention to label-specific information should be
ingrained into the mechanics of the classifier, decoupling it from semantic information. This becomes even
more important in a few-shot setting where the network has to quickly learn from little data." I do not see a proof of this concept in the empirical results. Where is the proof that when label-specific information and semantic information are coupled, then performance is worse and when what the authors propose is implemented, performance improves. There is no ablation study like this in the paper and moreover, the section "Ablation study" is doing hyperparameter search instead of doing the ablation study. In ablation study we would like to see the effects of adding and removing key components of the proposed architecture, as well observing any interaction effects between them. Please fix this.
- Another claim in the paper is the $z_s$ learns semantic representation of an image. Please provide empirical evidence supporting this claim. At the moment it is not at all clear what exactly learned in $z_s$ and why it helps (if indeed it does) to learn a better $z_l$.
- Please prove that $z_s$ helps to learn better $z_l$. What if we do not supervise $z_s$  at all, may be we still get the same performance level?


**Strengths And Weaknesses:**

Strengths

- Interesting idea
- relevant topic
- paper is clearly written and easy to follow
- SOTA results

Weaknesses

- The effectiveness of the key ingredient of the idea, i.e. jointly learning representations of label and semantics, is not supported with empirical evidence
- No qualitative examples demonstrating the proposed intuition in action
- a few unsubstantiated claims
- ablation studies are vacuous

---

> ### Author Response · Authors · 2023-01-23
> **Response to Reviewer sHKr - Part 1**
>
> Thanks for your thought-provoking reviews and feedback on our paper! We are pleased to know that you find our ideas interesting and our paper easy to follow. In the following, we will address your concerns, questions and recommendations in a point-to-point fashion.
> > Q1: I do not understand what decoupled stands for ....
>
> A1: Thanks for this question. The proposed approach is called ``decoupled" since we infer two pivotal characteristics of an image by modelling them as two \textit{separate} latent variables - $\textbf{z}_l$ and $\textbf{z}_s$. We model these two latent variables independent of each other, as reflected in the generative process of $\texttt{TRIDENT}$ (equation 1, Figure 2). However, $\textbf{z}_l$ depends on $\textbf{z}_s$ only during inference due to the presence of a collider node $\textbf{x}$ [1]. This does not change the fact that we model these two latent variables unconditionally independent of each other. By the way, this matter is clearly articulated in the caption of Figure 2. Notably, Figure 2 shows a dotted arrow from $\textbf{z}_s$ to $\textbf{z}_l$ indicating dependence in inference and not generation.
> > Q2: Equation deriving ELBO uses factorization ...
>
> A2: Thanks for your reflection on the idea and associated theoretical derivations. We refer to all class-characterising attributes of an image (such as wings of a bird, hump on a camel’s back) as `"label" information ($\textbf{z}_l$) and all the $\textit{other}$ different attributes $\textbf{that are not necessarily relevant for its classification}$ (for example: context, background, style, design) as "semantic" information ($\textbf{z}_l$), as explained in Section 1. Building on this definition, the two latent variables are modelled to be independent of each other since they encode/account for two entirely different and separate pieces of information.
> Secondly, we respectfully disagree with your comment on the contradiction of Figure 1 with the theoretical foundation of our paper. Figure 1 shows an arrow from $\textbf{z}_s$ to $\textbf{z}_l$ signifying the dependence of $\textbf{z}_l$ on $\textbf{z}_s$, $\textbf{only during inference}$ of $\textbf{z}_l$. As already explained in Subsection 4.2, we must include $\textbf{z}_s$ in the inference of $\textbf{z}_l$ due to the presence of a collider node $\textbf{x}$ [1]. This collider node $\textbf{x}$ between $\textbf{z}_l$ and $\textbf{z}_s$ makes the two parent latent variables unconditionally independent; however, conditionally dependent given $\textbf{x}$ [1]. As a result, the unconditional independence of $\textbf{z}_l$ and $\textbf{z}_s$ implies our generative process’ factorisation $p(\textbf{z}_l, \textbf{z}_s) = p(\textbf{z}_l) p(\textbf{z}_s)$ and the conditional dependence given $\textbf{x}$ implies our inference formulation $q(\textbf{z}_l, \textbf{z}_s) = q_\{\phi_\{1\}\}(\textbf{z}_l \mid \textbf{z}_s, \textbf{x}) q_\{\phi_\{2\}\}(\textbf{z}_s \mid \textbf{x})$.
> > Q3: Related work fails to demonstrate...
>
> A3: Thanks for your attention to this. We actually have an extensive review of the current literature (roughly covering $40$ relevant studies) in Section 1 following which Section 2 (Related Work) indeed dives deep into different categories of approaches. Following your suggestion, and to further clarify this connection, we have added extra explanations to Section 2 to revisit how $\texttt{TRIDENT}$ places itself against the current state-of-the-art.
> > Q4: Relationship and novelty w.r.t. matching...
>
> A4: Thanks for the comment. We already discuss the novel aspects of $\texttt{TRIDENT}$ in Sections 1 and 2. Following your suggestion, we have now included a more specific discussion on the novelty w.r.t. Matching Networks and other attention-based few-shot learning methods in Section 4.3, which states "Unlike other attention-based few-shot learning methods (Ye et al., 2020, Vinyals et al., 2016), we do not compute an attention-based transform on the flattened support and query vectors, but rather on the outputs of the $\texttt{Conv}_\{1\times1\}(.; \textbf{W}_N)$ to effectively fuse information from multiple class-pixel comparisons".

---

> ### Author Response · Authors · 2023-01-23
> **Response to Reviewer sHKr - Part 2**
>
> > Q5: "This helps the network utilize information across......
>
> A5: Thanks for the comment. We included this statement as an intuitive explanation of our design choices in $\texttt{AttFEX}$ regarding a 1$\times$1 convolution, and not as a claim. Since the 1$\times$1 convolution is computed across feature maps of all images in a task, a weighted sum/difference of all class pixels is computed implicitly. As the weights over class pixels are decided by the parameters of the 1$\times$1 convolution $\textbf{W}_M$, we chose to call it a "parametric" comparison of classes. Following your suggestion, we have actually tried to visualise the feature maps output by our $\texttt{AttFEX}$ module. However, given the fact that the feature map $\textbf{F} = \tt{ConvEnc}(\textbf{X})$ is reduced to a much smaller size of $H' \times W' = 5 \times 5$, the visualizations are not very indicative of a class-wise comparison. Nonetheless, we have modified this statement in Subsection 4.3 as follows: "This helps the network utilize information across corresponding pixels of all images in a task $\mathcal{T}_i$, which can be considered as a parametric comparison of classes.".
> > Q6: "This increases the sensitivity of the...
>
> A6: Thanks for this question. The claim regarding the "sensitivity of network parameters" during MAML-style training is originally borrowed from Finn et al., 2017 where meta-learning using MAML was initially proposed. Other works such as How to train your MAML [2], FO-MAML (Nichol et al., 2018), Implicit gradient MAML (Rajeswaran et al., 2018) and BOIL (Oh et al., 2021) also discuss this claim and build on it. Following your suggestion, we further elaborate on this in Subsection 4.5, Training Strategy and refer interested readers to the demonstration in the corresponding references.
> > Q7: The main idea and claim of the paper is "Thus, we ...
>
> A7: Thanks for the comment. We are a bit surprised why the already-existing experimentation on these very exact topics is overlooked! Let us further elaborate on this in the following. We have already tested this exact experimentation in Table 7, Subsection 5.4 as $\texttt{ConvFEX}$, where the label and semantic information are not modelled separately as $\textbf{z}_l$ and $\textbf{z}_s$, respectively. In $\texttt{ConvFEX}$, only a single latent variable $\textbf{z}$ is learnt to perform few-shot classification; thus, making it responsible for both label-specific and semantic information. The results corroborate that when these two pieces of information are coupled we observe a significant performance degradation as compared to $\texttt{TRIDENT}$, thus reaffirming the importance of our $\textit{decoupled}$ variational inference strategy.
>
> **Response** to *"the section Ablation study is doing hyperparameter search.."*: We agree that the ablation study should not be focused on hyperparameter search but a carefully thought out study of the various components of $\texttt{TRIDENT}$ - which by our best efforts is already the case in Subsection 5.3. More concretely, varying the meta-batch size $B$ and inner-update steps $n$ indicates the impact of meta-learning on our proposed approach, testing different values of the dimensionality of $\textbf{z}_l$, $\textbf{z}_s$ indicate the dependence of $\texttt{TRIDENT}$ on the compression capacity of the encoder, and varying the number of channels in $\textbf{W}_N$, $\textbf{W}_M$ indicate how sensitive $\texttt{TRIDENT}$ is to the different number of pixel-wise comparisons in $\texttt{AttFEX}$. Additionally, we also study the effects of removing key components of the architecture in Subsection 5.4, Table 7, where we drop our transductive feature extractor $\texttt{AttFEX}$ and the variational inference branch of $\textbf{z}_s$. Following your suggestion, as well as the suggestion provided by **Reviewer MEaT**, we will now include further ablation studies and analyses of various components as depicted in ResponseTables 2,3,4 and 5. That said, we are happy to add further ablation experimentation per your suggestion.

---

> ### Author Response · Authors · 2023-01-23
> **Response to Reviewer sHKr - Part 3**
>
> > Q8: Another claim in the paper is the $z_s$ learns.... **and** Please prove that $z_s$ helps to learn ....
>
> A8: As already explained in Section 1, all the $\textit{other}$ different attributes that are not necessarily relevant for an image's classification (for example- context, background, style, design) are defined as "semantic" information ($\textbf{z}_s$) for the purposes of our research. To empirically show that $\textbf{z}_s$ contains classification-irrelevant information, we conduct the analysis shown in Figure 5, Subsection 5.3 where the drop in DBI score of $\textbf{z}_s$ after the meta-update is significantly smaller than the drop in the case of $\textbf{z}_l$ ($2.461$ vs. $6.723$). This indicates that the class separation in $\textbf{z}_s$ after the meta update is much worse than that of $\textbf{z}_l$, thus corroborating our claim that $\textbf{z}_s$ encodes class-irrelevant information. We called this class-irrelevant information "semantic" information, even though after all the core idea is that whatever that information is, it must be refactored from class-informative context (in $\textbf{z}_l$) to provide superior performance. Please note that this is already illustrated in Table 7, in the experimentation on $\texttt{ConvFEX}$ where only a single latent variable is learnt (practically merging $\textbf{z}_l$, $\textbf{z}_s$ into one). The drop in performance of $\texttt{ConvFEX}$ as compared to $\texttt{TRIDENT}$ empirically verifies the importance of learning $\textbf{z}_s$ and our decoupling strategy.
>
> **References**:
>
> [1] Glymour, Madelyn, Judea Pearl, and Nicholas P. Jewell. Causal inference in statistics: A primer. John Wiley \& Sons, 2016.
>
> [2] Antoniou, Antreas, Harrison Edwards, and Amos Storkey. "How to train your MAML." arXiv preprint arXiv:1810.09502 (2018).

---

### Author Response · Authors · 2023-01-24
**Response to all reviewers - Manuscript Revision**

We thank all the reviewers for their thorough and constructive feedback.

Based on the suggestions and actionable feedback received from all the reviewers, we have revised our manuscript accordingly. The revisions and additions have been coloured $\textcolor{blue}{blue}$.

Please let us know in case you have any further questions and we would be happy to respond to them. We hope that our revisions are satisfactory and sufficient in addressing all your remarks.

Regards,

Authors

---

### Decision · Action_Editors · 2023-02-08

**Recommendation:** Accept with minor revision

**Comment:**

This paper presents a model for few-shot classification that makes use of two latent representations of an image, each with a different semantic content (one for context information, one for label information). The paper also introduces an attention-based network to extract features that exploits information from both query and support images of the few-shot task. The effectiveness of the proposed model and network, trained with variational inference, is demonstrated through a thorough set of experiments.

In the initial version of the paper, the reviewers raised some valid concerns, including: clarity/writing issues (including some mistakes in the equation), missing ablation studies, choice of baselines, and incomplete citations. This was mostly fixed by the authors through an extensive revision, in line with the reviewers' suggestions.

After the authors' rebuttal, Reviewers MEaT and Y8UK recognized that the paper had been significantly improved and that their concerns had been addressed. Reviewer sHKr, however, still has some concerns and leans towards rejection. Specifically:
+ The concerns about "decoupling" and "inference" seem to be caused by a mere nomenclature misunderstanding. Here, "decoupling" refers to being able to recover two latent representations, each with different semantic content. Perhaps a less confusing way of stating that would be dropping the word "decouple" and simply states that the model considers both types of latent variables, but I think it is also fine to leave as-is. The term "inference" refers to the process of learning the latent variables through approximate posterior inference (as opposed to "at test time"). In this sense, the model and inference are allowed to be different, and any choice of the variational model will lead to a valid ELBO - this should also address the point about the "theoretical justification".
+ Regarding the *missing citations*, the authors have now added citations to original work in Section 4.5, as well as a connection to matching networks in Section 4.3.
+ The *ablation analysis* in Table 7 shows the importance of the proposed architecture, while Section 5.4 (last paragraph) also analyzes the importance of having two decoupled latent variables.
+ *Figure 5* shows empirically how $z_l$ captures label information whereas $z_c$ doesn't. As a **suggestion to the authors**, another way to quantify that would be to compute some mutual information between $Z_l$ and the labels (and between $Z_c$ and the labels); it may be nice to include that in the text as well.

I thus recommend acceptance. In the discussion period, the reviewers also mentioned the paper can be improved as there are some remianing writing issues. I encourage the authors to take a full pass over the text and also to take care of any of the suggestions from the reviewers that haven't been incorporated yet.

Finally, a couple of minor comments:
+ Please replace "Fig. 5.2" with "Fig. 5".
+ Please add a cross-link in the last paragraph of Section 5.4.

**Audience:**

The topic of few-shot learning is of interest of the TMLR's audience, and so are the paper contributions in terms of modeling, architecture, and empirical findings. Specifically, the approach performs very well on common benchmarks.

**Claims And Evidence:**

Overall, I find the extensive experiments added during the rebuttal period show empirical evidence towards the model that considers both types of latent variables separately, together with the proposed architecture. See specific comments below.